# On the Power of Small-size Graph Neural Networks for Linear Programming

**Qian Li[1,2,*], Tian Ding[1,2,*], Linxin Yang[2,3,*], Minghui Ouyang[4], Qingjiang Shi[5], Ruoyu Sun[1,2,3,†]**

[1] Shenzhen International Center For Industrial And Applied Mathematics, Shenzhen, China
[2] Shenzhen Research Institute of Big Data, Shenzhen, China
[3] School of Data Science, The Chinese University of Hong Kong, Shenzhen, China
[4] Peking University, Beijing, China
[5] Tongji University, Shanghai, China

## Abstract

Graph neural networks (GNNs) have recently emerged as powerful tools for addressing complex optimization problems. It has been theoretically demonstrated that GNNs can universally approximate the solution mapping functions of linear programming (LP) problems. However, these theoretical results typically require GNNs to have large parameter sizes. Conversely, empirical experiments have shown that relatively small GNNs can solve LPs effectively, revealing a significant discrepancy between theoretical predictions and practical observations. In this work, we aim to bridge this gap by providing a theoretical foundation for the effectiveness of smaller GNNs. We prove that polylogarithmic-depth, constant-width GNNs are sufficient to solve packing and covering LPs, two widely used classes of LPs. Our proof leverages the capability of GNNs to simulate a variant of the gradient descent algorithm on a carefully selected potential function. Additionally, we introduce a new GNN architecture, termed GD-Net. Experimental results demonstrate that GD-Net significantly outperforms conventional GNN structures while using fewer parameters.

## 1 Introduction

Learning to Optimize (L2O) has emerged as a compelling research area, leveraging machine learning techniques to enhance the efficiency of optimization processes. Unlike traditional theory-driven optimization methods, L2O approaches are primarily data-driven, learning optimization strategies from existing problem instances. This new paradigm has yielded notable advancements in both continuous optimization [22] and combinatorial optimization [4, 21].

Recently, graph neural networks (GNN) have become increasingly popular in the field of L2O [6, 27]. GNNs are a class of neural networks specifically designed to process and analyze data structured as graphs, leveraging the relationships between nodes to perform tasks such as node classification, link prediction, and graph classification [35]. Due to their properties of permutation equivariance and natural adaptation to varying input dimensions, GNNs are well-suited for graph-related optimization problems such as minimum vertex covering [30] and traveling salesman [13]. Recent research has demonstrated that GNNs can effectively accelerate the solving process for both linear programming (LP) and mixed integer linear programming (MILP) problems, which are among the most important and widely applied types of optimization problems. For instance, Li et al. [17] proposed a GNN-based reformulation method for LP to enhance the solver performance. Furthermore, Chen et al. [8] and

---

[*]These authors contributed equally to this work.

[†]Corresponding Author. Email:sunruoyu@cuhk.edu.cn

38th Conference on Neural Information Processing Systems (NeurIPS 2024).

Qian et al. [29] explored the potential of GNNs to approximate the solution mapping of LP and reported encouraging results. Additionally, several studies have proposed various approaches to guide MILP solvers with GNNs [10, 11, 14, 18, 26, 32].

Despite the numerical success, the theoretical foundation of using GNN to solve optimization problems remains less clear. Initial steps towards a theoretical understanding have been made by Chen et al. [8] and Qian et al. [29]. They demonstrated that with a sufficient number of parameters, GNNs can approximate the solution mapping of LPs with arbitrary precision. However, a significant gap persists between the theoretical progress and empirical evidence. The proof in [8] relies heavily on the universal approximation theorem for multi-layer perceptions (MLP), which necessitates a large number of parameters in principle. Similarly, the result of [29] requires the depth of GNN to be polynomial in the problem dimension. In practice, however, GNN with a modest width and fewer than ten layers often suffice to achieve good performance in approximating the optimal solution of LP with hundreds of nodes and constraints.

This gap between theory and practice raises an intriguing open question:

*When and why can small-size GNNs effectively solve LPs?*

As larger networks usually require more training examples and higher computational resources, addressing this question is not only of theoretical interest but also provides potential guidance to L2O practitioners. While current GNNs may not yet rival theoretically grounded LP solvers in precision, they could still accelerate LP solving by providing high-quality initializations to warm-start traditional solvers. Understanding the principles behind the success of small-sized GNNs can potentially lead to more parameter-efficient models, thus reducing the computational resources required for LP solving. Moreover, these insights could potentially enhance MILP solvers. Specifically, the leading approach for MILP is the branch-and-bound framework, which involves selecting variables and dividing the search space. A prevalent variable selection approach is strong branching, where variables are scored by solving associated LPs to guide the branching decisions. Researchers have exploited GNNs to score these variables, with GNNs essentially solving the associated LPs [9–11, 23, 31]. By uncovering the mechanism behind the success of small-sized GNNs in solving LPs, these models could be further improved, leading to more efficient MILP solving.

## 1.1 Our contribution

In this paper, we show that *polylogarithmic-depth constant-width* GNNs can approximate the solution mapping for a broad class of LPs, namely packing LPs and covering LPs, with arbitrary precision (Theorem 3 and Theorem 5). This result advances previous theoretical results on general LPs [8, 29], narrowing the gap between existing theoretical progress and empirical evidence.

**Packing and covering LPs and their norm form.** A *packing LP* and its dual *covering LP* are nonnegative LPs of the canonical form: $\max\{\boldsymbol{c}^T \cdot \boldsymbol{x} \mid \boldsymbol{A}\boldsymbol{x} \leq \boldsymbol{b}, \boldsymbol{x} \geq 0\}$ and $\min\{\boldsymbol{b}^T \cdot \boldsymbol{y} \mid \boldsymbol{A}^T \boldsymbol{y} \geq \boldsymbol{c}, \boldsymbol{y} \geq 0\}$ respectively. Here, $\boldsymbol{A} \in \mathbb{R}_{\geq 0}^{n \times m}$, $\boldsymbol{b} \in \mathbb{R}_{\geq 0}^{n}$, and $\boldsymbol{c} \in \mathbb{R}_{\geq 0}^{m}$. Packing LPs and covering LPs are broad classes of linear programming problems that can be used to approximate or relax a wide range of fundamental problems in combinatorial optimization. They include fractional versions of the vertex cover problem, the set cover problem, the hypergraph matching problem, the dominating set problem, and the maximum independent set problem. Without loss of generality, we assume the packing and covering LPs are presented in their *normal form* [2], where the vectors $\boldsymbol{b}$ and $\boldsymbol{c}$ are all-ones vectors and $A_{ij}$ is either zero or greater than 1. The details of the reduction to the normal form can be found in the appendix.

**Our results.** We present a new theoretical explanation of the phenomenon that small-size GNNs can solve general packing LPs and covering LPs, by unrolling gradient descent algorithms. Specifically, we first propose a variant of the gradient descent (GD) algorithm proposed by Awerbuch and Khandekar [2] for solving packing and covering LPs (Algorithms 1 and 2), so that our variant can be more naturally simulated by GNNs, where one iteration of the algorithm can be simulated by one GNN layer of constant width. Importantly, our variant of GD is guaranteed to output a $(1 + \epsilon)$-approximate solution in $\mathrm{polylog}(mnA_{\max}/\epsilon)$ iterations. Therefore, we affirm the feasibility of solving general packing and covering LPs with *polylogarithmic-depth constant-width* GNNs. Here, we remark that the polylogarithmic dependency on depth is also necessary. Specifically, Kuhn et

al. [16] showed that: for the fractional maximum matching problem, a special kind of packing LP, every constant-factor approximation distributed algorithm requires at least $\Omega(\sqrt{\log n / \log \log n})$ rounds. Moreover, since one layer of GNNs can be naturally simulated by one round of distributed LP algorithms (see, e.g., the second paragraph on page 5 in [16]), we conclude that GNNs need at least $\Omega(\sqrt{\log n / \log \log n})$ layers.

Building on our theoretical proof, we propose a new GNN architecture, termed GD-Net, for solving packing and covering LPs. We empirically demonstrate that, when appropriately trained, GD-Net can effectively adapt to given problem instance distributions. Experiments across various datasets demonstrate that GD-Net outperforms classical graph convolutional network (GCN), a predominant GNN architecture in L2O for LP and MILP solving. Specifically, GD-Net generates better solutions with an order of magnitude fewer parameters than GCN. These performance enhancements become increasingly significant as the problem dimensions expand. The numerical success of GD-Net not only corroborates our theoretical framework but also suggests a potential direction for designing more parameter-efficient GNNs for L2O. This direction involves developing architectures that can simulate well-established, theoretically grounded algorithms, potentially leading to improvements in both computational efficiency and solution quality.

## 1.2 Related works

The design of GD-Net is based on the concept of unrolling iterative algorithms as GNNs. Indeed, there is a body of research that has explored this approach. For example, Velickovic et al. [33] investigated solving basic graph problems (e.g., the shortest path, the minimum spanning tree) by GNNs. By unrolling classical graph algorithms (e.g., breadth-first search, Prim's algorithm) as GNNs, they suggest that message-passing neural networks with a maximization aggregator may be best suited for such graph problems. Aiming at mitigating oversmoothing, long-range dependencies, and spurious edges issues of GNNs, Yang et al. [36] proposed a new family of GNN layers by unrolling and integrating the update rules of two classical iterative algorithms, namely the proximal gradient descent and iterative reweighted least squares. Papers [20, 38] showed that many existing GNN models (such as GCN, GAT, APPNP) can be viewed as unrolling gradient descent serving specific graph signal denoising problems. Chen et al. [7] proposed new GNNs to improve graph signal denoising by unrolling sparse coding and trend filtering algorithms. Papers [24, 39, 37] bridge the gap between graph convolution and iterative algorithms by providing a unified optimization framework for GNNs.

## 2 Gradient descent algorithm for packing and covering LPs

Throughout the paper, we use boldface to represent vectors or matrices, e.g., $\boldsymbol{x}$ and $\boldsymbol{A}$. Let $\mathbf{1}_{p \times q}$ denote the $p \times q$-dimensional all-ones matrix, and $\mathbf{1}_p$ denote the $p$-dimensional all-ones column vector. Similarly, we can define $\mathbf{0}_{p \times q}$ and $\mathbf{0}_p$.

**Algorithm for packing LPs.** Awerbuch and Khandekar [2] proposed a simple $(1+\epsilon)$-approximation algorithm for packing LPs of the normal form:

$$\max\{\mathbf{1}^T \cdot \boldsymbol{x} \mid \boldsymbol{A}\boldsymbol{x} \leq \mathbf{1}, \boldsymbol{x} \geq \mathbf{0}\} \tag{1}$$

where $A_{ij}, b_i, c_j$ are all non-negative and $A_{ij}$ is either 0 or $\geq 1$. The algorithm is described as follows: given parameters $\mu = \frac{1}{\epsilon} \cdot \ln \frac{m A_{\max}}{\epsilon}$, $\alpha = \frac{\epsilon}{4}$, $\beta = \frac{\alpha}{10\mu}$, and $\delta = \frac{\alpha}{10\mu n A_{\max}}$;

– Start with $\boldsymbol{x} := \mathbf{0}_m$; Then, repeat the following procedure:

1. $y_i := \exp[\mu \cdot (\boldsymbol{A}_i \boldsymbol{x} - 1)]$ for each $i \in [n]$;

2. For any $j \in [m]$ do

    (a) If $\boldsymbol{A}_j^T \boldsymbol{y} \leq 1 - \alpha$, then $x_j := \max\{x_j \cdot (1 + \beta), \delta\}$;

    (b) If $\boldsymbol{A}_j^T \boldsymbol{y} \geq 1 + \alpha$, then $x_j := x_j \cdot (1 - \beta)$.

---

**Algorithm 1:** Our variant of Awerbuch-Khandekar algorithm for packing LPs

---

1  **Input**: A $n \times m$ matrix $\boldsymbol{A}$ where $A_{ij}$ is either zero or no less than 1, and $\epsilon > 0$.

2  **Parameter:** $\mu := \frac{1}{\epsilon} \cdot \ln \frac{mA_{\max}}{\epsilon}$, $\alpha = \frac{\epsilon}{4}$, $\beta = \frac{\alpha}{10\mu}$, and $\delta = \frac{\alpha}{10\mu n A_{\max}}$;

3  Initialize $\boldsymbol{x}^0 := \boldsymbol{0}$;

4  **for** $k = 0$ *to* $K - 1$ **do**

5     $y_i^k := \exp[\mu(\boldsymbol{A}_i \boldsymbol{x}^k - 1)]$ for any $i \in [n]$;

6     **for** *each* $j \in [m]$ **do**

7         **if** $\boldsymbol{A}_j^T \boldsymbol{y}^k \leq 1 - \alpha$ **then** $x_j^{k+1} := x_j^k \cdot (1 + \beta) + \delta$;

8         **if** $\boldsymbol{A}_j^T \boldsymbol{y}^k \geq 1 + \alpha$ **then** $x_j^{k+1} := x_j^k \cdot (1 - \beta)$;

9  **Output**: $\boldsymbol{x}^K$.

---

The intuition behind this algorithm is that it can be viewed as applying gradient descent on a carefully chosen potential function defined as

$$\Phi_p(\boldsymbol{x}) = \sum_{i=1}^{n} \frac{y_i(\boldsymbol{x})}{\mu} - \sum_{j=1}^{m} x_j = \sum_{i=1}^{n} \frac{\exp[\mu(A_i \boldsymbol{x} - 1)]}{\mu} - \sum_{j=1}^{m} x_j,$$

which enjoys the following nice properties [2]: (i) $\Phi_p(\boldsymbol{x})$ is differentiable and convex; (ii) $\nabla \Phi_p(\boldsymbol{x}) = \boldsymbol{A}^T \boldsymbol{y}$; and (iii) any stationary point of $\Phi_p$ is a nearly optimal solution to the packing LP (1): For any $\boldsymbol{x}$ where $\nabla \Phi_p(\boldsymbol{x}) = 0$, $\boldsymbol{x}$ is feasible and $\boldsymbol{1}^T \cdot \boldsymbol{x} \geq (1 - 4\epsilon) \cdot \text{OPT}$. It turns out that this algorithm has *polylogarithmic convergence* [2]. Precisely, given any $\epsilon > 0$, this algorithm always maintains a feasible solution (i.e. $\boldsymbol{x} \geq 0$ and $\boldsymbol{Ax} \leq \boldsymbol{1}$ always hold), and returns a $(1 + \epsilon)$-approximation solution to the packing LP (1) in $\tilde{O}(\ln^2(mA_{max}) \cdot \ln^2(nA_{\max})/\epsilon^5)$ iterations. Here $\tilde{O}$ hides lower order terms like $\ln \ln(mn A_{\max})$ and $\ln(1/\epsilon)$.

**Algorithm for covering LPs.** In the same paper [2], Awerbuch and Khandekar also considered solving the dual covering LP of the normal form

$$\min\{\boldsymbol{1}^T \cdot \boldsymbol{y} \mid \boldsymbol{A}^T \boldsymbol{y} \geq \boldsymbol{1}, \boldsymbol{y} \geq 0\}, \tag{2}$$

and designed the following algorithm: given $\mu := \frac{1}{\epsilon} \cdot \ln \frac{nA_{\max}}{\epsilon}$, $\alpha = \frac{\epsilon}{4}$, $\beta = \frac{\alpha}{20\mu}$, and $\delta = \frac{\alpha}{20\mu m A_{\max}}$,

– Start with $\boldsymbol{y} := \boldsymbol{1}_n$; Then repeat the following procedure:

1. $x_j := \exp[\mu \cdot (1 - \boldsymbol{A}_j^T \boldsymbol{y})]$ for each $j \in [m]$;

2. For any $i \in [n]$ do
   (a) If $\boldsymbol{A}_i \boldsymbol{x} \geq 1 + \alpha$, then $y_i := \max\{y_i \cdot (1 + \beta), \delta\}$;
   (b) If $\boldsymbol{A}_i \boldsymbol{x} \leq 1 - \alpha$, then $y_i := y_i \cdot (1 - \beta)$.

Similarly, this algorithm can also be thought of as applying a variant of gradient descent on a carefully selected potential function [2]:

$$\Phi_c(\boldsymbol{y}) = \sum_{j=1}^{m} \frac{x_j(\boldsymbol{y})}{\mu} + \sum_{i=1}^{n} y_i = \sum_{j=1}^{m} \frac{\exp[\mu(1 - \boldsymbol{A}_j^T \boldsymbol{y})]}{\mu} + \sum_{i=1}^{n} y_i,$$

which satisfies that: (i) $\Phi_c(\boldsymbol{y})$ is differentiable and convex; (ii) $\nabla \Phi_c(\boldsymbol{y}) = \boldsymbol{1} - \boldsymbol{Ax}$; and (iii) for any $\boldsymbol{y}$ where $\nabla \Phi_d(\boldsymbol{y}) = 0$, $\boldsymbol{y}$ is feasible and $\boldsymbol{1}^T \cdot \boldsymbol{y} \leq (1 + 4\epsilon) \cdot \text{OPT}$. This algorithm was shown to enjoy polylogarithmic convergence as well [2]. Formally, this algorithm always maintains a feasible solution (i.e. $\boldsymbol{y} \geq 0$ and $\boldsymbol{A}^T \boldsymbol{y} \geq \boldsymbol{1}$ always hold), and returns a $(1 + \epsilon)$-approximation solution to the covering LP (2) in $\tilde{O}(\ln^2(nA_{max}) \cdot \ln^2(mn A_{\max})/\epsilon^5)$ iterations.

**A variant of Awerbuch-Khandekar algorithms.** We propose a variant of the Awerbuch-Khandekar algorithms, specifically Algorithm 1 for solving packing LPs (1) and Algorithm 2 for covering LPs (2). The motivation and advantage of our variant is that it can be simulated more naturally by GNNs.

---

**Algorithm 2:** Our variant of Awerbuch-Khandekar algorithm for covering LPs

---

1 **Input**: A $n \times m$ matrix $\boldsymbol{A}$ where $A_{ij}$ is either zero or no less than 1, and $\epsilon > 0$.

2 **Parameter:** $\mu := \frac{1}{\epsilon} \cdot \ln \frac{nA_{\max}}{\epsilon}$, $\alpha = \frac{\epsilon}{4}$, $\beta = \frac{\alpha}{20\mu}$, and $\delta = \frac{\alpha}{20\mu m A_{\max}}$;

3 Initialize $\boldsymbol{y}^0 := \mathbf{1}$;

4 **for** $k = 0$ *to* $K - 1$ **do**

5     $x_j^k := \exp[\mu(1 - \boldsymbol{A}_j^T \boldsymbol{y}^k)]$ for any $j \in [m]$;

6     **for** *each* $i \in [n]$ **do**

7        **if** $\boldsymbol{A}_i \boldsymbol{x}^k \geq 1 + \alpha$ **then** $y_i^{k+1} := y_i^k \cdot (1 + \beta) + \delta$ ;

8        **if** $\boldsymbol{A}_i \boldsymbol{x}^k \leq 1 - \alpha$ **then** $y_i^{k+1} := y_i^k \cdot (1 - \beta)$ ;

9 **Output**: $\boldsymbol{y}^K$.

---

Basically, we replace $x_j \leftarrow \max\{x_j(1 + \beta), \delta\}$ with $x_j \leftarrow x_j(1 + \beta) + \delta$ in their algorithm for packing LPs, and $y_j \leftarrow \max\{y_j(1 + \beta), \delta\}$ with $y_j \leftarrow y_j(1 + \beta) + \delta$ in their algorithm for covering LPs. Through almost the same proof, one can verify that the polylogarithmic convergence still holds.

**Theorem 1.** *Algorithm 1 always maintains a feasible solution and returns a $(1 + \epsilon)$-approximation solution to the packing LP (1) in $\tilde{O}(\ln^2(mA_{max}) \cdot \ln^2(nA_{\max})/\epsilon^5)$ iterations.*

*Algorithm 2 always maintains a feasible solution and returns a $(1 + \epsilon)$-approximation solution to the covering LP (2) in $\tilde{O}(\ln^2(nA_{max}) \cdot \ln^2(mnA_{\max})/\epsilon^5)$ iterations.*

**Connection to GNN.** For LPs, an instance can be naturally encoded as a labeled bipartite graph [1, 3, 15, 25], where (a) a left node represents a variable, (b) a right node represents a constraint (and equivalently, the associated dual variable), and (c) a left node and a right node are connected if the corresponding variable participates the corresponding constraint. In our GD algorithms, the matrix-vector multiplication $\mathbf{Ax}$ (or $\mathbf{A}^T \mathbf{y}$) can be interpreted as a message passing step from left nodes to right nodes (or from right nodes to left nodes). In the next sections, we will demonstrate how one iteration of our GD algorithms can be transformed into a single layer of GNN.

## 3 Design of packing GD-Net

By combining the idea of Algorithm 1 and techniques from graph neural networks, we design a graph neural network architecture, named packing GD-Net, for solving packing LPs.

One iteration of Algorithm 1 consists of two steps: first, the algorithm updates $\boldsymbol{y}$ from $\boldsymbol{x}$, and calculates the gradient $\nabla\Phi_p(\boldsymbol{x}) = \boldsymbol{A}^T \boldsymbol{y} - \mathbf{1}$; second, using $\nabla\Phi_p(\boldsymbol{x})$, it applies a variant of gradient descent to update $\boldsymbol{x}$. Our packing GD-Net modifies the second step by replacing it with a learnable neural network block, while leaving the first step unchanged. Thus, the packing GD-Net can also be viewed as utilizing a neural network block to accelerate the convergence of the gradient descent.

**ELU activation for $\boldsymbol{y}$-update.** We apply the Exponential Linear Unit (ELU) activation function to replicate the $\boldsymbol{y}$-update: $y_i^k = \exp[\mu(\boldsymbol{A}_i \boldsymbol{x}^k - 1)]$. We will fix the parameter $\alpha$ in ELU to 1, then $\text{ELU}(t) + 1 = \exp(t)$ for $t \leq 0$. In addition, as mentioned in Theorem 1, $\boldsymbol{A}_i \boldsymbol{x}^k - 1 \leq 0$ always holds in the execution of Algorithm 1. Thus the ELU function can exactly replicate the $\boldsymbol{y}$-update. Specifically, our packing GD-Net updates $\boldsymbol{y}$ as follows:

$$y_i^k := \text{ELU}\big[\mu(\boldsymbol{A}_i \boldsymbol{x}^k - 1)\big] + 1. \tag{3}$$

Since $\alpha$ is fixed to 1 and $\mu$ is fixed to $\frac{1}{\epsilon} \cdot \ln \frac{mA_{\max}}{\epsilon}$, no learnable parameters are involved here.

**Learnable gradient descent procedure.** The gradient descent in Algorithm 1 can be rewritten as

$$x_j^{k+1} := x_j^k + f\left(\boldsymbol{A}_j^T \boldsymbol{y}^k - 1\right) \cdot x_j^k + g\left(\boldsymbol{A}_j^T \boldsymbol{y}^k - 1\right) = x_j^k + f\left(\partial\Phi_p/\partial x_j\right) \cdot x_j^k + g\left(\partial\Phi_p/\partial x_j\right)$$

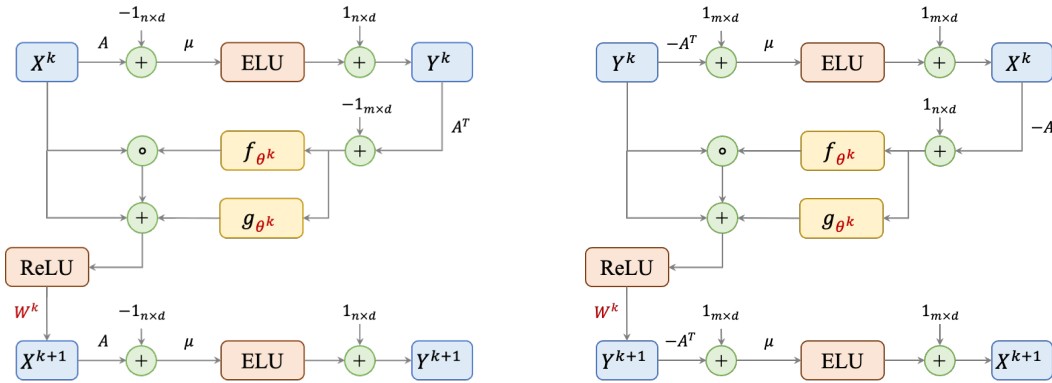

Figure 1: The architectures of a single layer in packing (left) and covering (right) GD-Nets. Learnable parameters are colored in red.

where $f, g : \mathbb{R} \to \mathbb{R}$ are sums of Heaviside step functions defined as:

$$f(t) = \begin{cases} \beta, & \text{if } t \leq -\alpha; \\ 0, & \text{if } -\alpha \leq t \leq \alpha; \quad \text{and} \quad g(t) = \begin{cases} \delta, & \text{if } t \leq -\alpha; \\ 0, & \text{otherwise.} \end{cases} \\ -\beta, & \text{if } t \geq \alpha. \end{cases} \tag{4}$$

As the Heaviside step function can be naturally approximated by the sigmoid function $\sigma(t) = \frac{1}{1+\exp(-t)}$, the packing GD-Net adopts the following learnable functions as substitutes for $f$ and $g$:

$$f_{\boldsymbol{\theta}^k}(t) = \sum_{\ell=1}^{L_1} \left[ \theta_{\ell,1}^k \cdot \sigma(\theta_{\ell,2}^k \cdot t + \theta_{\ell,3}^k) + \theta_{\ell,1}^k \cdot \sigma(\theta_{\ell,2}^k \cdot t - \theta_{\ell,3}^k) - \theta_{\ell,1}^k \right] \tag{5}$$

and

$$g_{\boldsymbol{\theta}^k}(t) = \sum_{\ell=1}^{L_2} \theta_{\ell,4}^k \cdot \left[ \sigma(\theta_{\ell,5}^k \cdot t + \theta_{\ell,6}^k) - \sigma(\theta_{\ell,6}^k) \right]. \tag{6}$$

Note that it is guaranteed that $f_{\boldsymbol{\theta}^k}(t) = -f_{\boldsymbol{\theta}^k}(-t)$, in particular $f_{\boldsymbol{\theta}^k}(0) = 0$, and $g_{\boldsymbol{\theta}^k}(0) = 0$. So if the packing GD-Net reaches a stationary point of $\Phi_p$, it will remain there. Additionally, the packing GD-Net applies the ReLU activation to keep $x_j^k \geq 0$.

**Channel expansion and architecture of packing GD-Net**  We integrate the channel expansion technique to further strengthen the expressive power. Specifically, we expand the $m$-dimensional column vector $\boldsymbol{x}^k$ into a $(m \times d)$-dimensional matrix $\boldsymbol{X}^k$, and $n$-dimensional column vector $\boldsymbol{y}^k$ into a $(n \times d)$-dimensional matrix $\boldsymbol{Y}^k$. Now, we are ready to present the packing GD-Net architecture:

– Initialize $\boldsymbol{X}^0 := \mathbf{0}_{m \times d}$ and $\boldsymbol{Y}^0 = \mathbf{0}_{n \times d}$.

– For $k = 0, 1, 2, \cdots, K-1$

- $\boldsymbol{Y}^k := \text{ELU}\left[\mu(\boldsymbol{A}_i \boldsymbol{X}^k - \mathbf{1}_{n \times d})\right] + \mathbf{1}_{n \times d}$.

- $\boldsymbol{X}^{k+1} = \text{ReLU}\left( \left\{ \boldsymbol{X}^k + f_{\boldsymbol{\theta}^k}\left[\boldsymbol{A}^T \boldsymbol{Y}^k - \mathbf{1}_{m \times d}\right] \circ \boldsymbol{X}^k + g_{\boldsymbol{\theta}^k}\left[\boldsymbol{A}^T \boldsymbol{Y}^k - \mathbf{1}_{m \times d}\right] \right\} \cdot \boldsymbol{W}^k \right)$.

– Output $\boldsymbol{x}^{final} := \boldsymbol{X}^K \cdot \boldsymbol{w}^K \in \mathbb{R}^m$.

Here, $\circ$ denotes entry-wise multiplication (a.k.a. Hadamard product), and $f_{\boldsymbol{\theta}^k}, g_{\boldsymbol{\theta}^k} : \mathbb{R} \to \mathbb{R}$ are applied entrywise to the matrix $\boldsymbol{A}^T \boldsymbol{Y}^k - \mathbf{1}_{m \times d}$. The learnable parameter is $\boldsymbol{\Theta} := \{\boldsymbol{\theta}^k, \boldsymbol{W}^k\}_{k=0}^{K-1} \cup \{\boldsymbol{w}^K\}$, where $\boldsymbol{\theta}^k = \{\theta_{\ell,1}^k, \theta_{\ell,2}^k, \theta_{\ell,3}^k\}_{\ell=1}^{L_1} \cup \{\theta_{\ell,4}^k, \theta_{\ell,5}^k, \theta_{\ell,6}^k\}_{\ell=1}^{L_2}$, $\boldsymbol{W}^k \in \mathbb{R}^{d \times d}$, and $\boldsymbol{w}^K \in \mathbb{R}^{d \times 1}$. So the total number of parameters is $K \cdot (d^2 + 3L_1 + 3L_2) + d$. The following theorem reveals the ability of the packing GD-Net to reproduce Algorithm 1 with arbitrarily small precision.

**Theorem 2.** *Given any network depth $K$, any network width $d \geq 1$, and any precison $\eta > 0$, there exists a $K$-layer packing GD-Net with a specific choice of parameters such that, for any packing LP instance (1), $\|\boldsymbol{x}^{final} - \boldsymbol{x}^{alg}\| \leq \eta$. Here, $\boldsymbol{x}^{alg}$ is the output of $K$-iteration Algorithm 1.*

By combining Theorems 1 and 2, we conclude that the packing GD-Nets with polylogarithmic depth and constant width are sufficient to solve packing LPs.

**Theorem 3.** *Given any $\epsilon > 0$ and $\eta > 0$, there exists a packing GD-Net of $\tilde{O}(\ln^2(mA_{max}) \cdot \ln^2(nA_{\max})/\epsilon^5)$ depth and constant width, using the parameter assignment $\boldsymbol{\Theta}_{\mathrm{GD}}$, that satisfies the following condition: For any packing LP instance (1), $\boldsymbol{x}^{final}$ is $\eta$-close to being a $(1+\epsilon)$-approximate solution.*

**Remark 1.** *We can establish a similar theorem with general-purpose GNNs (such as GCNs) where we can still derive polylogarithmic depth in the bound but constant width is no longer guaranteed. Specifically, by the universal approximation theorem, the ELU function can be simulated with arbitrary precision by a 2-layer and sufficiently wide perceptron. So if we replace each occurrence of ELU with this 2-layer perceptron, we then obtain a GCNs. Note that this GCN still has polylogarithmic depth but the required width is no longer a constant.*

**Network training.** The training data set is a set $\mathcal{I} = \{(\boldsymbol{A}, \boldsymbol{x}^*)\}$ of packing LP instances in the normal form. More specifically, the input of an instance is identified by the constraint matrix $\boldsymbol{A}$, with $\boldsymbol{b}$ and $\boldsymbol{c}$ being both all-ones vectors; the label $\boldsymbol{x}^*$ represents the corresponding optimal solution. Let $\boldsymbol{x}^{final}(\boldsymbol{\Theta}, \boldsymbol{A})$ denote the output of the packing GD-Net parameterized by $\boldsymbol{\Theta}$ running on the input $\boldsymbol{A}$. The goal of the training process is to find a parameter $\boldsymbol{\Theta}^*$ minimizing loss function defined as:

$$\mathcal{L}_p(\mathcal{I}; \boldsymbol{\Theta}) = \frac{1}{|\mathcal{I}|} \sum_{(\boldsymbol{A}, \boldsymbol{x}^*) \in \mathcal{I}} \|\boldsymbol{x}^{final}(\boldsymbol{\Theta}, \boldsymbol{A}) - \boldsymbol{x}^*\|^2.$$

**Feasibility resortation.** Note that the $\boldsymbol{x}^{final}$ returned by packing GD-Net may be infeasible. To restore feasibility, we implement the following post-processing procedure:

– First, for each $j \in [m]$, update $x_j := \max(0, \min(1, x_j))$;

– Then, for $i = 1$ to $n$ do

  • If $\boldsymbol{A}_i \boldsymbol{x} \geq 1$, then update $x_j := \frac{x_j}{\boldsymbol{A}_i \boldsymbol{x}}$ for each $j$ with $A_{ij} \neq 0$.

## 4 Design of covering GD-Net

In one iteration of Algorithm 2, the process begins by updating $\boldsymbol{x}$ from $\boldsymbol{y}$, followed by the calculation of the gradient $\nabla \Phi_d(\boldsymbol{y}) = \mathbf{1} - \boldsymbol{A}\boldsymbol{x}$. Finally, it applies a variant of gradient descent to update $\boldsymbol{y}$. Our covering GD-Net retains the original $\boldsymbol{x}$-update and gradient-calculation modules, but replaces the gradient descent with a learnable neural network block to accelerate convergence.

**Architecture of covering GD-Net.** Similarly, we substitute the ELU activation with $\alpha$ fixed to 1 for the $\exp(\cdot)$ function, and replace the $\boldsymbol{x}$-update $x_j^k = \exp[\mu(1 - \boldsymbol{A}_j^T \boldsymbol{y}^k)]$ with

$$x_j^k := \mathrm{ELU}\big[\mu(1 - \boldsymbol{A}_j^T \boldsymbol{y}^k)\big] + 1. \tag{7}$$

As we will show in Theorem 4, (7) can exactly simulate the $\boldsymbol{x}$-update in Algorithm 2.

The gradient descent procedure in Algorithm 2 can be rewritten as

$$y_i^{k+1} := y_i^k + f\big(1 - \boldsymbol{A}_i \boldsymbol{x}^k\big) \cdot y_i^k + g\big(1 - \boldsymbol{A}_i \boldsymbol{x}^k\big) = y_i^k + f\big(\partial \Phi_d / \partial y_i\big) \cdot y_i^k + g\big(\partial \Phi_d / \partial y_i\big)$$

where $f, g : \mathbb{R} \to \mathbb{R}$ are the same functions as those defined in (4) for packing GD-Net. So, in the covering GD-Net, we also substitute $f$ and $g$ with the learnable functions $f_{\boldsymbol{\theta}^k}$ and $g_{\boldsymbol{\theta}^k}$ defined in (5) and (6) respectively. In addition, we also apply ReLU to keep $y_i^k \geq 0$.

Besides, the channel expansion technique is also incorporated: similarly, we expand $\boldsymbol{x}^k \in \mathbb{R}^m$ into $\boldsymbol{X}^k \in \mathbb{R}^{m \times d}$, and $\boldsymbol{y}^k \in \mathbb{R}^n$ into $\boldsymbol{Y}^k \in \mathbb{R}^{n \times d}$. Then, we propose our covering GD-Net architecture:

– Initialize $\boldsymbol{Y}^0 := \mathbf{1}_{n \times d}$ and $\boldsymbol{X}^0 = \mathbf{1}_{m \times d}$.

– For $k = 0, 1, 2, \cdots, K - 1$

- $\boldsymbol{X}^k := \mathrm{ELU}\big[\mu(\mathbf{1}_{m \times d} - \boldsymbol{A}^T \boldsymbol{Y}^k)\big] + \mathbf{1}_{m \times d}$;

- $\boldsymbol{Y}^{k+1} = \mathrm{ReLU}\left(\left\{\boldsymbol{Y}^k + f_{\boldsymbol{\theta}^k}\big[\mathbf{1}_{n \times d} - \boldsymbol{A}\boldsymbol{X}^k\big] \circ \boldsymbol{Y}^k + g_{\boldsymbol{\theta}^k}\big[\mathbf{1}_{n \times d} - \boldsymbol{A}\boldsymbol{X}^k\big]\right\} \cdot \boldsymbol{W}^k\right).$

– Output $\boldsymbol{y}^{final} := \boldsymbol{Y}^K \cdot \boldsymbol{w}^K \in \mathbb{R}^n$.

The learnable parameter is $\boldsymbol{\Theta} := \{\boldsymbol{\theta}^k, \boldsymbol{W}^k\}_{k=0}^{K-1} \cup \{\boldsymbol{w}^K\}$, where $\boldsymbol{\theta}^k = \{\theta_{\ell,1}^k, \theta_{\ell,2}^k, \theta_{\ell,3}^k\}_{\ell=1}^{L_1} \cup \{\theta_{\ell,4}^k, \theta_{\ell,5}^k, \theta_{\ell,6}^k\}_{\ell=1}^{L_2}$, $\boldsymbol{W}^k \in \mathbb{R}^{d \times d}$, and $\boldsymbol{w}^K \in \mathbb{R}^{d \times 1}$. So the total number of parameters is $K \cdot (d^2 + 3L_1 + 3L_2) + d$, the same as in packing GD-Net.

**Theorem 4.** *Given any network depth $K$, any network width $d \geq 1$, and any precison $\eta > 0$, there exists a $K$-layer covering GD-Net with a specific choice of parameters such that, for any covering LP instance (2), $\|\boldsymbol{y}^{final} - \boldsymbol{y}^{alg}\| \leq \eta$. Here, $\boldsymbol{y}^{alg}$ is the output of $K$-iteration Algorithm 2.*

Theorems 1 and 4 together imply the capacity of polylogarithmic-depth constant-width covering GD-Nets for solving covering LPs.

**Theorem 5.** *Given any $\epsilon > 0$ and $\eta > 0$, there exists a covering GD-Net of $\tilde{O}(\ln^2(nA_{max}) \cdot \ln^2(mnA_{\max})/\epsilon^5)$ depth and constant width, using the parameter assignment $\boldsymbol{\Theta}_{\mathrm{GD}}$, that satisfies the following condition: For any covering LP instance (2), $\boldsymbol{y}^{final}$ is $\eta$-close to being a $(1 + \epsilon)$-approximate solution.*

**Remark 2.** *Similar to Remark 1, we can establish a similar theorem with general-purpose GNNs where the polylogarithmic bound on the depth still holds but constant width is no longer guaranteed.*

**Network training.** The training dataset is a set $\mathcal{I} = (\boldsymbol{A}, \boldsymbol{y}^*)$ consisting of instances where $\boldsymbol{y}^*$ is the optimal solution to the covering LP (2) with constraint matrix $\boldsymbol{A}$. The goal of the training process is to find a parameter $\boldsymbol{\Theta}^*$ minimizing the following loss function:

$$\mathcal{L}_c(\mathcal{I}; \boldsymbol{\Theta}) := \frac{1}{|\mathcal{I}|} \sum_{(A, \boldsymbol{y}^*) \in \mathcal{I}} \|\boldsymbol{y}^{final}(\boldsymbol{\Theta}, \boldsymbol{A}) - \boldsymbol{y}^*\|_2^2,$$

**Feasibility resortation.** Since the $\boldsymbol{y}^{final}$ may be infeasible, we implement the following post-processing procedure to restore feasibility:

– First, for each $i \in [n]$, update $y_i := \max(0, \min(1, y_i))$;

– Then, for $j = 1$ to $m$ do

  • If $\boldsymbol{A}_j^T \boldsymbol{y} \leq 1$, then update $y_i := \frac{y_i}{\boldsymbol{A}_j^T \boldsymbol{y}}$ for each $i$ with $A_{ij} \neq 0$.

# 5 Experimental study

## 5.1 Experemental Setup

**Datasets.** We utilized four LP relaxations of publicly available mixed-integer optimization instances as benchmarks. Specifically, we included the Maximal Independent Set (IS), Packing Problem (Packing), Edge Covering Problem (ECP), and Set Covering (SC). The problem definitions are adopted from [9, 29]. Each benchmark comprises four sets of problem instances with varying sizes, including one set designated for the generalization experiment. Detailed information regarding the problem sizes and data splitting ratios can be found in the appendix.

To construct datasets for training, each instance $M_i \equiv \{\boldsymbol{A}^i, \boldsymbol{b}^i, \boldsymbol{c}^i\}$ undergoes normalization to $\widetilde{M}_i \equiv \{\hat{\boldsymbol{A}}^i, \mathbf{1}_n, \mathbf{1}_m\}$. Subsequently, via the optimization solver SCIP [5], we obtain the optimal solution and optimal objective value pair $\{\boldsymbol{x}^i, \mathrm{obj}_i^*\}$ for each instance $\widetilde{M}_i$. Finally, we utilize the input-target pair to compose the dataset $\mathcal{D} \equiv \{\hat{\boldsymbol{A}}^i, \boldsymbol{x}^i\}_{i=1}^{|M|}$.

**Models and Training settings.** For comparison, our experiments also include the graph convolutional network (GCN), a predominant GNN architecture in L2O for LP and MILP. Specifically,

we adopt the GCN implementation from [29], which is tailored for predicting the optimal solutions for LPs. Both the GCN and the proposed GD-Net utilize a four-layer architecture with 64 hidden units in each layer. Consequently, the number of parameters is 1,656 for GD-Net and 34,306 for GCN. Note that our GD-Net has an order of magnitude fewer parameters compared to GCN. For GD-Net, we set $\epsilon = 0.2$. All models were trained using a learning rate of $10^{-3}$. We trained each model for 10,000 epochs, and the checkpoint with the lowest validation loss was saved for evaluation. For reproducibility, our code to conduct the experiments can be found at `https://anonymous.4open.science/r/GD-Net-6FC7/`.

**Metrics.** To effectively evaluate the GNN's ability to solve LP problems, we employed two distinct metrics: the relative gap R. Gap $= |\widetilde{obj}_i - obj_i^*|/obj_i^*$ and the absolute gap (A. Gap $= |\widetilde{obj}_i - obj_i^*|$), where $\widetilde{obj}_i$ denotes the predicted objective value of the respective approach after feasibility restoration.

## 5.2 Comparing against GCNs

In this section, we assess the effectiveness of the proposed GD-Net and the GCNs adopted from [29] in predicting the optimal solution for LP problems. As shown in Table 1, we report relative and absolute gaps, along with validation and test errors, to evaluate the quality of solutions generated by each model. The results indicate that GD-Net typically achieves narrower gaps compared

Table 1: Results of comparing the proposed GD-Net against GCNs from [29]. We report valid/test errors measured by MSE (V.Err/T.Err) and the relative/absolute objective gap from the optimal solution (R. Gap/A.Gap). Better performances are highlighted in bold. Results are averaged across 100 instances.

| Dataset | Size | GD-Net | | | | GCNs [29] | | | |
|---------|------|--------|--------|--------|--------|-----------|--------|--------|--------|
| | | V. Err | T. Err | R. Gap | A. Gap | V. Err | T. Err | R. Gap | A. Gap |
| IS | S | 0.062 | 0.062 | **4.41**% | 1.478 | 0.145 | 0.122 | 15.46% | 5.155 |
| | M | 0.156 | 0.135 | **3.55**% | 12.201 | 0.156 | 0.135 | 16.18% | 55.318 |
| | L | 0.085 | 0.085 | **6.84**% | 43.073 | 0.154 | 0.131 | 15.35% | 96.785 |
| Packing | S | 3.40E-4 | 3.39E-4 | **16.53**% | 0.184 | 2.7E-4 | 2.6E-4 | 19.50% | 0.220 |
| | M | 6.53E-4 | 6.53E-4 | **10.68**% | 0.118 | 5.05E-4 | 5.09E-4 | 10.69% | 0.118 |
| | L | 2.18E-4 | 2.20E-4 | **7.35**% | 0.082 | 1.69E-4 | 1.69E-4 | 7.37% | 0.082 |
| ECP | S | 0.099 | 0.097 | **7.84**% | 1.478 | 0.173 | 0.153 | 36.83% | 12.41 |
| | M | 0.129 | 0.115 | **21.51**% | 74.80 | 0.172 | 0.153 | 38.73% | 134.50 |
| | L | 0.123 | 0.115 | **18.28**% | 116.05 | 0.172 | 0.153 | 39.22% | 249.06 |
| SC | S | 3.12E-4 | 2.91E-4 | 26.68% | 0.297 | 2.53E-4 | 2.54E-4 | **21.91**% | 0.244 |
| | M | 6.64E-6 | 6.50E-6 | 13.09% | 0.145 | 5.05E-6 | 5.10E-6 | **10.91**% | 0.121 |
| | L | 2.19E-6 | 2.19E-6 | **8.42**% | 0.094 | 1.81E-6 | 1.77E-6 | 8.90% | 0.099 |

to GCNs. Even in instances where GD-Net does not surpass GCNs, the performance discrepancy remains minimal, which is noteworthy given GD-Net's significantly fewer parameters. Notably, in scenarios like `Packing-L`, although GD-Net records a higher test error, it still outperforms GCNs. This suggests that GD-Net may better capture the structural nuances of the problem. Overall, GD-Net generally demonstrates superior performance over conventional GCNs.

## 5.3 Generalization to larger instances

In this section, we evaluate the generalization capability of our proposed GD-Net, trained on smaller instances, to larger problem domains. We train GD-Nets on the largest dataset available for each problem and subsequently test these models on problem instances with $10\%$ more constraints and variables. Table 2 presents the results, showcasing relative and absolute gaps, along with validation and test errors. The results reveal that GD-Nets possess a notable ability to generalize to larger problem instances with only minimal performance degradation. This suggests robustness in handling increased problem complexity, underscoring the adaptability and scalability of the proposed GD-Nets.

Table 2: Results of generalizing GD-Nets trained on smaller instances to larger instances. All models are trained on datasets of size L. We report valid/test errors measured by MSE (V.Err/T.Err) and the relative/absolute objective gap from the optimal solution (R. Gap/A.Gap). Results are averaged across 100 instances.

| dataset | n | m | V. Err | T. Err | R. Gap | A. Gap |
|---------|---|---|--------|--------|--------|--------|
| IS | [1100, 1300] | [1100, 1300] | 0.085 | 0.085 | 6.81% | 47.681 |
| Packing | [1100, 1300] | [1100, 1300] | 2.18E-4 | 1.85E-6 | 7.06% | 0.078 |
| ECP | [1100, 1300] | [1100, 1300] | 0.123 | 0.115 | 17.48% | 120.63 |
| SC | [1100, 1300] | [1100, 1300] | 2.19E-6 | 1.87E-6 | 10.68% | 0.119 |

## 5.4 Comparing against more Baselines

To further demonstrate the effectiveness of the proposed framework, we include two additional baselines: the traditional first-order solver PDLP [19] and the commercial solver Gurobi [12]. Specifically, we used Gurobi's primal simplex method, which efficiently produces feasible primal solutions. The table below shows the time taken by each method to achieve solutions with the same precision level as GD-Net's solutions. The experiment was conducted on both the SC and Packing datasets across all three size variants.

Table 3: Performance comparison of GD-Net, Gurobi, and PDLP on achieving the same precision.

| Instance | #Vars. | Optimal Obj. | GD-Net Obj. | GD-Net Time | Gurobi Time | PDLP Time |
|----------|--------|--------------|-------------|-------------|-------------|-----------|
| | 1,000 | 3.334 | 3.701 | 0.105s | 0.244s | 0.919s |
| SC | 5,000 | 100.667 | 130.931 | 0.218s | 9.401s | 0.921s |
| | 10,000 | 407.386 | 546.666 | 0.335s | 103.322s | 1.001s |
| | 1,000 | 3.334 | 3.018 | 0.095s | 0.208s | 0.746s |
| Packing | 5,000 | 100.88 | 78.994 | 0.216s | 3.980s | 0.756s |
| | 10,000 | 406.946 | 302.04 | 0.314s | 8.593s | 0.809s |

As shown in Table 3, GD-Net consistently outperforms both general-purpose solvers on all datasets. Notably, PDLP, a first-order method known for its fast early-stage convergence, is unable to produce solutions of comparable quality to those of GD-Net in shorter time. This further highlights GD-Net's ability to efficiently generate high-quality solutions. Additionally, as previously mentioned, the simplex method requires matrix factorization, which is computationally expensive. In this case, it required up to $300\times$ more time to converge to a solution of the same quality as GD-Net. These findings strongly support the effectiveness of GD-Net, demonstrating its capability to consistently generate high-quality solutions.

Moreover, we also include experiments under a more practical setting and compare the inference time, which can be found in Appendix G and H.

## 6 Conclusion

Inspired by Awerbuch and Khandekar's gradient descent algorithms for packing and covering LPs, we introduce packing and covering GD-Net, and prove that they can approximate the solution mapping of packing and covering LPs respectively. Importantly, they only need polylogarithmic depth and constant width, significantly narrowing the gap between existing theoretical prediction and empirical evidence. Experiments are also conducted to demonstrate their effectiveness. We list some directions for future work: (1) How to further reduce the size of GNNs theoretically, since there is still a gap between our theoretical progress and empirical evidence; (2) How low the size of GNNs can go for solving general LPs, noting that our nets only work for packing and covering LPs; (3) To explore our nets in L2O for MILP. Recall that GD-Nets can be viewed as unrolling the gradient descent on a carefully selected potential function with some good properties. For (2) and (3), a natural direction is to design other potential functions that still enjoy those good properties.

## Acknowledgments

This paper is supported by the National Key Research and Development Project under grant 2022YFA1003900; Hetao Shenzhen-Hong Kong Science and Technology Innovation Cooperation Zone Project (No.HZQSWS-KCCYB-2024016); University Development Fund UDF01001491, the Chinese University of Hong Kong, Shenzhen; Guangdong Provincial Key Laboratory of Mathematical Foundations for Artificial Intelligence (2023B1212010001); the Guangdong Major Project of Basic and Applied Basic Research (2023B0303000001). Qian Li and Minghui Ouyang's work was additionally supported by the National Natural Science Foundation of China Grants No.62002229. Tian Ding's work was additionally supported by the Internal Project of Shenzhen Research Institute of Big Data under Grant J00220240005.

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

# A    Packing and covering LP: reduction to the normal form

Recall that a packing LP and its dual covering LP are nonnegative LPs of the canonical form:

$$\max_{\boldsymbol{x}} \quad \boldsymbol{c}^T \boldsymbol{x} \qquad \text{(Packing LP)} \qquad\qquad \min_{\boldsymbol{y}} \quad \boldsymbol{b}^T \boldsymbol{y} \qquad \text{(Covering LP)}$$

$$\text{s.t.} \quad \boldsymbol{A}\boldsymbol{x} \leq \boldsymbol{b} \qquad\qquad\qquad\qquad \text{s.t.} \quad \boldsymbol{A}^T \boldsymbol{y} \geq \boldsymbol{c}$$
$$\boldsymbol{x} \geq 0 \qquad\qquad\qquad\qquad\qquad\qquad \boldsymbol{y} \geq 0$$

where all $A_{ij}, b_i$, and $c_j$ are non-negative. Their norm forms are those where $\boldsymbol{b}$ and $\boldsymbol{c}$ are both all-ones vectors and $A_{ij}$ is either zero or greater than 1.

$$\max_{\boldsymbol{x}} \quad \mathbf{1}^T \boldsymbol{x} \qquad\qquad\qquad\qquad \min_{\boldsymbol{y}} \quad \mathbf{1}^T \boldsymbol{y}$$

$$\text{s.t.} \quad \boldsymbol{A}\boldsymbol{x} \leq \mathbf{1} \qquad\qquad\qquad\qquad \text{s.t.} \quad \boldsymbol{A}^T \boldsymbol{y} \geq \mathbf{1}$$
$$\boldsymbol{x} \geq 0 \qquad\qquad\qquad\qquad\qquad\qquad \boldsymbol{y} \geq 0$$

The reduction to the normal form proceeds as follows. First, we can assume that each $b_i > 0$ since otherwise all variables $x_j$ with $A_{ij} > 0$ have to be zero and $y_i$ can be set to zero; similarly, we can assume each $c_j > 0$ since otherwise $x_j$ can be set to zero and all variables $y_i$ with $A_{ij} > 0$ have to be zero. Then, we can replace $A_{ij}$ by $\hat{A}_{ij} = \frac{A_{ij}}{b_i c_j}$, replace $\boldsymbol{b}$ and $\boldsymbol{c}$ by all-ones vector, and work with variables $\hat{x}_j = c_j x_j$ and $\hat{y}_i = b_i y_i$. Finally, we replace $\hat{A}_{ij}$ by $\tilde{A}_{ij} = \frac{\hat{A}_{ij}}{\min\{\hat{A}_{i'j'}|\hat{A}_{i'j'}>0\}}$ and work with $\tilde{x}_j = \hat{x}_j \cdot \min\{\hat{A}_{i'j'} \mid \hat{A}_{i'j'}>0\}$ and $\tilde{y}_i = \hat{y}_i \cdot \min\{\hat{A}_{i'j'} \mid \hat{A}_{i'j'}>0\}$.

# B    Proof of Theorem 2

*Proof.* As increasing the network width $d$ does not decrease the expressive power, it suffices to show the theorem for $d = 1$. When $d = 1$ and $\boldsymbol{w}^K = 1$, then packing GD-Net reduces to

– Initialize $\boldsymbol{x}^0 := \mathbf{0}_m$ and $\boldsymbol{y}^0 = \mathbf{0}_n$.

– For $k = 0, 1, 2, \cdots, K-1$

- $y_i^k := \text{ELU}\big[\mu(\boldsymbol{A}_i \boldsymbol{x}^k - 1)\big] + 1$ for each $i \in [n]$;

- $x_j^{k+1} := \text{ReLU}\left[x_j^k + f_{\boldsymbol{\theta}^k}\left(\boldsymbol{A}_j^T \boldsymbol{y} - 1\right) \cdot x_j^k + g_{\boldsymbol{\theta}^k}\left(\boldsymbol{A}_j^T \boldsymbol{y} - 1\right)\right]$ for each $j \in [m]$;

– Output $\boldsymbol{x}^{final} := \boldsymbol{x}^K$.

For the $\boldsymbol{y}$-update part, recalling that in the execution of Algorithm 1, $\boldsymbol{x}$ is always feasible, thus $\boldsymbol{A}_i \boldsymbol{x} - 1 \leq 0$ (Theorem 1). Additionally, since $\alpha$ is fixed to 1, $\text{ELU}(t) + 1 = \exp[t]$ for $t \leq 0$. Therefore, in Algorithm 1, we can replace $y_i^k := \exp[\mu(\boldsymbol{A}_i \boldsymbol{x}^k - 1)]$ with $y_i^k := \text{ELU}\big[\mu(\boldsymbol{A}_i \boldsymbol{x}^k - 1)\big] + 1$ without altering the algorithm's behavior.

For the $\boldsymbol{x}$-update part, we rewrite this update in Algorithm 1 as

$$x_j^{k+1} := x_j^k + f\left(\boldsymbol{A}_j^T \boldsymbol{y} - 1\right) \cdot x_j^k + g\left(\boldsymbol{A}_j^T \boldsymbol{y} - 1\right),$$

where $f$ and $g$ are defined in (4). Both $f$ and $g$ can be expressed as the sum of at most two Heaviside step functions, which can be naturally simulated by a sigmoid function with arbitrarily small error. Specifically, by setting $L_1 = 1, \theta_{1,1} = -\beta, \theta_{1,3} = \alpha \cdot \theta_{1,2}$, and making $\theta_{1,2}^k$ sufficiently large, $f_{\boldsymbol{\theta}^k}$ can approximate $f$ with arbitrary precision. Similarly, by setting $L_2 = 1, \theta_{1,4}^k = -\delta, \theta_{1,6}^k = \alpha \cdot \theta_{1,5}^k$, and making $\theta_{1,5}^k$ large enough, $g$ can be approximated arbitrarily well by $g_{\boldsymbol{\theta}^k}$. Therefore, in Algorithm 1, if we substitute

$$x_j^{k+1} := x_j^k + f_{\boldsymbol{\theta}^k}\left(\boldsymbol{A}_j^T \boldsymbol{y} - 1\right) \cdot x_j^k + g_{\boldsymbol{\theta}^k}\left(\boldsymbol{A}_j^T \boldsymbol{y} - 1\right),$$

we will get the above reduced packing GD-Net, and the change of the output $\boldsymbol{x}^K$ can be made arbitrarily small by appropriately choosing the parameter $\boldsymbol{\Theta}$. $\qquad\square$

## C   Proof of Theorem 4

*Proof.* The proof is very similar to that of Theorem 2. First, we only need to show the theorem for $d = 1$, because it does not decrease the expressive power to increase the network width $d$. When $d = 1$, and by letting $\boldsymbol{w}^K = 1$, the covering GD-Net reduces to

– Initialize $\boldsymbol{y}^0 := \mathbf{1}_n$ and $\boldsymbol{x}^0 = \mathbf{1}_m$.

– For $k = 0, 1, 2, \cdots, K - 1$

- $x_j^k := \mathrm{ELU}\big[\mu(1 - \boldsymbol{A}_j^T \boldsymbol{y}^k)\big] + 1$ for each $j \in [m]$;

- $y_i^{k+1} := \mathrm{ReLU}\big[y_i^k + f_{\boldsymbol{\theta}^k}\left(1 - \boldsymbol{A}_i \boldsymbol{x}\right) \cdot y_i^k + g_{\boldsymbol{\theta}^k}\left(1 - \boldsymbol{A}_i \boldsymbol{x}\right)\big]$ for each $i \in [n]$;

– Output $\boldsymbol{y}^{final} := \boldsymbol{y}^K$.

For the $\boldsymbol{x}$-update part, recalling that in the execution of Algorithm 2, $\boldsymbol{y}$ is always feasible, thus $1 - \boldsymbol{A}_j^T \boldsymbol{y} \leq 0$ (Theorem 1). Additionally, since $\alpha$ is fixed to 1, $\mathrm{ELU}(t) + 1 = \exp[t]$ for $t \leq 0$. Therefore, in Algorithm 2, we can replace $x_j^k := \exp[\mu(1 - \boldsymbol{A}_j^T \boldsymbol{y}^k)]$ with $x_j^k := \mathrm{ELU}\big[\mu(1 - \boldsymbol{A}_j^T \boldsymbol{y}^k)\big] + 1$ without altering the algorithm's behavior.

For the $\boldsymbol{y}$-update part, we rewrite this update in Algorithm 1 as

$$y_i^{k+1} := y_i^k + f\left(1 - \boldsymbol{A}_i \boldsymbol{x}\right) \cdot y_i^k + g\big(1 - \boldsymbol{A}_i \boldsymbol{x}\big),$$

where $f$ and $g$ are defined in (4). As shown in the proof of Theorem 2, $f_{\boldsymbol{\theta}^k}$ and $g_{\boldsymbol{\theta}^k}$ can approximate $f$ and $g$ respectively with arbitrary precision. Therefore, in Algorithm 1, if we substitute

$$y_i^{k+1} := y_i^k + f_{\boldsymbol{\theta}^k}\left(1 - \boldsymbol{A}_i \boldsymbol{x}\right) \cdot y_i^k + g_{\boldsymbol{\theta}^k}\big(1 - \boldsymbol{A}_i \boldsymbol{x}\big),$$

then we will reach the above reduced covering GD-Net, and the change of the output $\boldsymbol{y}^K$ can be made arbitrarily small by appropriately choosing the parameter $\boldsymbol{\Theta}$. □

## D   Dataset specification

For all experiments, involved models are trained with datasets split into 5,000 training instances, 100 validation instances, and 100 test instances. Each problem contains 4 different sizes, namely small (S), medium (M), large (L), and generalization (Gen). The detailed sizes of each size can be found in Table 4-7.        For the IS benchmarks, we generated instances of specified sizes using the Ecole

Table 4: Sizes of Maximum Independent Set problems

| Size | # Row | # Col. |
|------|-------|--------|
| S | $[50 - 70]$ | $[50 - 70]$ |
| M | $[500 - 700]$ | $[500 - 700]$ |
| L | $[1000 - 1200]$ | $[1000 - 1200]$ |
| Gen | $[1100 - 1300]$ | $[1100 - 1300]$ |

Table 5: Sizes of Packing problems

| Size | # Row | # Col. | Density |
|------|-------|--------|---------|
| S | $[50 - 70]$ | $[50 - 70]$ | 60% |
| M | $[500 - 700]$ | $[500 - 700]$ | 60% |
| L | $[1000 - 1200]$ | $[1000 - 1200]$ | 60% |
| Gen | $[1100 - 1300]$ | $[1100 - 1300]$ | 60% |

library [28]. For ECP, we initially created IS instances, then converted these into their dual forms to obtain ECP instances. Regarding the SC and Packing datasets, we produced matrices of size $m \times n$ with a density of only $density\%$ non-zero entries. We then formulated the corresponding problems by transposing the matrix $A$ or leaving it as is, depending on the dataset requirements.

Table 6: Sizes of Edge Covering problems

| Size | # Row | # Col. |
|------|-------|--------|
| S | $[50 - 70]$ | $[50 - 70]$ |
| M | $[500 - 700]$ | $[500 - 700]$ |
| L | $[1000 - 1200]$ | $[1000 - 1200]$ |
| Gen | $[1100 - 1300]$ | $[1100 - 1300]$ |

Table 7: Sizes of Set Covering problems

| Size | # Row | # Col. | Density |
|------|-------|--------|---------|
| S | $[50 - 70]$ | $[50 - 70]$ | $60\%$ |
| M | $[500 - 700]$ | $[500 - 700]$ | $60\%$ |
| L | $[1000 - 1200]$ | $[1000 - 1200]$ | $60\%$ |
| Gen | $[1100 - 1300]$ | $[1100 - 1300]$ | $60\%$ |

## E  Hardware/Software specification

All experiments were performed on a server machine equipped with an Intel(R) Xeon(R) Platinum 8280 CPU @ 2.70GHz and 2.95 TB RAM. Data collection for training utilized Ecole 0.7.3 and Pyscipopt 4.2.0 for generating and solving instances. Model implementations were developed using PyTorch 2.1.

## F  Problem Definitions

In this section, we give the problem definitions and the specific formulations, as well as their LP relaxations of the original mixed-integer optimization problems used to generate data.

**Maximum Independent Set (IS).**   The goal is to select as many vertices as possible from an undirected graph $G = \{V, E\}$ such that, no two vertices form a edge.

$$\max_{\boldsymbol{x}} \quad \sum_{v \in V} x_v \quad \text{(IS)} \qquad\qquad \max_{\boldsymbol{x}} \quad \sum_{v \in V} x_v \quad \text{(LP relaxation)}$$
$$\text{s.t.} \quad x_u + x_v \leq 1, \forall (u,v) \in E \qquad\qquad \text{s.t.} \quad x_u + x_v \leq 1, \forall (u,v) \in E$$
$$\boldsymbol{x} \in \{0,1\}^{|V|} \qquad\qquad\qquad\qquad \boldsymbol{x} \geq \boldsymbol{0}$$

**Packing Problem.**   The goal is to maximize the profit from selecting from $m$ items, while the selected items must fit in $n$ resources constraints.

$$\max_{\boldsymbol{x}} \quad \sum_{j=1}^{m} c_j \cdot x_j \quad \text{(Packing)} \qquad\qquad \max_{\boldsymbol{x}} \quad \sum_{j=1}^{m} c_j \cdot x_j \quad \text{(LP relaxation)}$$
$$\text{s.t.} \quad \sum_{j=1}^{n} A_{ij} x_j \leq b_i, \forall i \in [1,n] \qquad\qquad \text{s.t.} \quad \sum_{j=1}^{n} A_{ij} x_j \leq b_i, \forall i \in [1,n]$$
$$\boldsymbol{x} \in \{0,1\}^{m} \qquad\qquad\qquad\qquad \boldsymbol{x} \geq \boldsymbol{0}$$

**Edge Covering Problem (SC).** This is the dual problem of the Maximum Independent Set problem, which minimizes the number of chosen edges such that every vertice touches at least one edge.

$$\min_{\boldsymbol{y}} \quad \sum_{e \in E} y_e \quad \text{(EC)} \qquad\qquad \min_{\boldsymbol{y}} \quad \sum_{e \in E} y_e \quad \text{(LP relaxation)}$$
$$\text{s.t.} \quad \sum_{e \ni v} y_e \geq 1, \forall v \qquad\qquad \text{s.t.} \quad \sum_{e \ni v} y_e \geq 1, \forall v$$
$$\boldsymbol{y} \in \{0,1\}^{E} \qquad\qquad\qquad\qquad \boldsymbol{y} \geq \boldsymbol{0}$$

**Set Covering (SC).** Given a family of subsets $S = \{s_1, \cdots, s_m\}$ where $s_i \subseteq [n]$, the goal is to select as few subsets as possible to cover all elements in $[n]$.

$$\min_{\boldsymbol{y}} \quad \sum_{s \in S} y_s \qquad \text{(SC)}$$
$$\text{s.t.} \quad \sum_{s \ni i} y_s \geq 1, \forall i \in [n]$$
$$\boldsymbol{y} \in \{0,1\}^S$$

$$\min_{\boldsymbol{y}} \quad \sum_{s \in S} y_s \qquad \text{(LP relaxation)}$$
$$\text{s.t.} \quad \sum_{s \ni i} y_s \geq 1, \forall i \in [n]$$
$$\boldsymbol{y} \geq \boldsymbol{0}$$

## G   Practical Problem

For the sake of more practical settings, we also included the Bipartite Maxflow problem (BMP) [34], which is a common model formulation applied to areas such as wireless communication. In our dataset, each bipartite graph is obtained by deleting all edges between $V'$ and $U'$ from a fully connected bipartite graph, and then randomly sample from the remaining edges with a probability of 60%, $V'$ (and $U'$ resp.) is a random subset consisting of half of the left nodes (and the right nodes). Based on the definition, we generated two sets of BMP problems with 1200 and 2000 nodes, respectively. In Table 8, we report the performance of GD-Net and compare it against GCNs.

Table 8: Comparison of GD-Net and GCN on BMP dataset

| #Nodes | GD-Net | | | GCNs | | |
|---|---|---|---|---|---|---|
| | Obj | A. Gap | R. Gap | Obj | A. Gap | R. Gap |
| 1200 | 35398.62 | 429.8 | **1.20%** | 31206 | 4622.41 | 12.89% |
| 2000 | 58844.8 | 943.33 | **1.58%** | 52085 | 7703.14 | 12.88% |

Based on the results, GD-Net consistently achieves better predictions compared to GCN, with an average of only a 1% optimality gap from the optimal solutions.

Additionally, we evaluated GD-Net against the conventional Ford-Fulkerson method, which is specifically designed for solving Maxflow problems. In Table 9, we report the time for Ford-Fulkerson to find solutions with the same quality of solutions predicted by GD-Net.

Table 9: Comparison of GD-Net and Ford-Fulkerson on different datasets

| #Nodes | GD-Net Obj | GD-Net Time | Ford-Fulkerson Time |
|---|---|---|---|
| 1200 | 35398.62 | **0.592s** | 2.152s |
| 2000 | 58844.80 | **1.691s** | 9.184s |

The results highlight that GD-Net is significantly faster than the Ford Fulkerson heuristic in achieving high-quality solutions, demonstrating its efficiency and effectiveness.

Table 10: Inference profiling of the proposed GD-Net and GCNs from [29]. Both the Inference time and the number of parameters are reported. All times are reported in seconds. Results are averaged across 100 instances.

| Model | # Parms. | IS | | | Packing | | | ECP | | | SC | | |
|---|---|---|---|---|---|---|---|---|---|---|---|---|---|
| | | S | M | L | S | M | L | S | M | L | S | M | L |
| GCNs [29] | 34,306 | 0.671 | 2.791 | 1.731 | 0.819 | 1.115 | 1.356 | 0.649 | 2.618 | 3.079 | 0.765 | 3.302 | 3.159 |
| GD-Net | 1,656 | 0.004 | 0.063 | 0.099 | 0.044 | 0.065 | 0.081 | 0.102 | 0.105 | 0.240 | 0.046 | 0.102 | 0.080 |

## H   Inference time profiling

To demonstrate the efficiency and scalability of GD-Nets, we present the average inference times of two models in Table 10. We see that GD-Nets consistently achieve faster inference times than

GCNs do. Furthermore, GD-Nets display strong scalability; their inference times remain comparably acceptable even as problem sizes increase. In contrast, GCNs require substantially more time to process and predict for larger problem instances.

