# OpenReview forum: "On the Power of Small-size Graph Neural Networks for Linear Programming"
_NeurIPS.cc/2024/Conference — NeurIPS 2024 poster_

### Official Review · Reviewer_h5Xh · 2024-06-24

**Soundness:** 3
**Presentation:** 3
**Contribution:** 3
**Rating:** 7
**Confidence:** 5

**Summary:**

This paper explores the effectiveness of small-sized GNNs in solving LP problems, addressing the discrepancy between theoretical predictions requiring large GNNs and empirical observations showing small GNNs' capability. The authors provide a theoretical foundation for the success of compact GNNs by proving that GNNs with polylogarithmic depth and constant width can solve specific LP classes, such as packing and covering LPs, by simulating a variant of the gradient descent algorithm. They introduce a novel GNN architecture, GD-Net, which outperforms traditional GNN models with fewer parameters. This understanding could lead to more efficient MILP solvers and reduce computational resources needed for solving LPs.

**Strengths:**

1. The paper successfully establishes a novel and intriguing link between the AK algorithm and GNNs. This connection is not only innovative but also has the potential to spark interest and pave the way for future research in the field.

2. The introduction of the unrolled GNN architecture represents a contribution to GNN-based LP and MILP solvers.

**Weaknesses:**

There are several areas where improvements could significantly enhance its contributions. Addressing these points satisfactorily would make a strong case for elevating the paper's status to "Accept".

1. The presentation of Theorems 2-4, given that GD-Net is an unrolled version of the AK algorithm, appears somewhat straightforward (Question 1,2).

2. To convincingly demonstrate the performance superiority of the proposed methods, it is crucial to include more realistic benchmarks and baselines for comparison (Question 3, 4).

3. Given that unrolled neural networks typically enjoy several advantages over more general neural architectures, it would be highly beneficial for the paper to explicitly verify these benefits in the context of the proposed GD-Net (Question 5).

**Questions:**

1. The paper claims that it closes the gap between theoretical and empirical perspectives in GNN-based LP solvers by illustrating the relationship between GD-Net and the AK algorithm. However, it has been observed that previous empirical studies predominantly utilized general-purpose GNNs, such as GCNs. Could similar theorems be achieved with general GNN backbones?

2. The demonstration that polylogarithmic-depth and constant-width GNNs can approximately solve LP problems is compelling. To better understand the significance of this result, could the authors discuss how close these findings are to the theoretical lower bound of what is required to solve LP problems?

3. This paper relaxes MILP problems to LPs and uses them as benchmarks, which may not fully reflect the practical significance of the proposed method. I suggest adopting more standard LP benchmarks and comparing GD-Net with heuristic algorithms.

4. In the specific setting described by [23], it is noted that GCNs underperform compared to GEN. To provide a comprehensive evaluation landscape, would it be possible for the authors to include GEN as a baseline in Table 1? This addition would help readers better assess the relative performance enhancements introduced by GD-Net.

5. The paper highlights the unique advantage of unrolling methods in achieving exceptional size generalization. However, the current presentation in Table 2 might not fully capture this advantage. Would the authors consider augmenting their experimental setup to include:
- Training on medium-scale datasets and testing on both small and large-scale datasets;
- Introducing the performance of GCNs as baselines for a direct comparison;

**Limitations:**

Yes

---

> ### Author Rebuttal · Authors · 2024-08-07
>
> > **Q1: It has been observed that previous empirical studies predominantly utilized general-purpose GNNs, such as GCNs. Could similar theorems be achieved with general GNN backbones?**
>
> A1: Yes, we can establish similar theorems with general-purpose GNNs. We can still derive polylog-depth in the bound, but constant width is no longer guaranteed.
>
> By the universal approximation theorem, the ELU activation function can be simulated with arbitrary precision by a 2-layer and sufficiently wide perceptron. So if we replace each occurrence of ELU in GD-net with this 2-layer perceptron, we then obtain a GCN. Note that this GCN still has polylog-depth but the required width is no longer constant in $\epsilon$.
>
> Thanks for the comment. We will add this discussion in the next version.
>
> > **Q2: Could the authors discuss how close these findings are to the theoretical lower bound of what is required to solve LP problems?**
>
> A2: The polylogarithmic depth is also the lower bound, and thus our bound is tight. Due to the space limit, we kindly refer Reviewer to **Part 5: Tightness of Our Bound** in "Author Rebuttal" for further elaboration. Thanks for the great comment. We will add this lower bound in the next version.
>
> > **Q3: This paper relaxes MILP problems to LPs and uses them as benchmarks, which may not fully reflect the practical significance of the proposed method. I suggest adopting more standard LP benchmarks and comparing GD-Net with heuristic algorithms.**
>
> A3:  We consider the Bipartite Maxflow, a common model formulation applied to areas such as wireless communication [R1]. In our dataset, each bipartite graph is obtained by deleting all edges between $V'$ and $U'$ from a fully connected bipartite graph, and then randomly sample from the remaining edges with a probability of 60%, where $V'$ (and $U'$ resp.) is a random subset consisting of half of the left nodes (and the right nodes).
>
> |        |          | GD-Net |           |       | GCN     |        |
> | ------ | -------- | ------ | --------- | ----- | ------- | ------ |
> | #Nodes | Obj      | A. Gap | R. Gap    | Obj   | A. Gap  | R. Gap |
> | 1200   | 35398.62 | 429.8  | **1.20%** | 31206 | 4622.41 | 12.89% |
> | 2000   | 58844.8  | 943.33 | **1.58%** | 52085 | 7703.14 | 12.88% |
> ||||||||
>
> According to the result, GD-Net consistently obtains better prediction compared to GCN, with only 1% optimality gap from optimal solutions.
>
> We also conducted comparisons against the Ford Fulkerson heuristic, specifically designed for solving maximum network flow problems. The table below presents the time taken to achieve the same optimality to GD-Net:
>
> | #Nodes | GD-Net Obj | GD-Net time | Ford Fulkerson time |
> | -: | -: | -: | -: |
> |1200 |35398.62 |0.592s |2.152s |
> |2000 |58844.80 |1.691s |9.184s |
> |||||
>
> GD-Net is significantly faster than the Ford Fulkerson heuristic in achieving high-quality solutions, demonstrating its efficiency and effectiveness.
>
> > **Q4: In the specific setting described by [23], it is noted that GCNs underperform compared to GEN. To provide a comprehensive evaluation landscape, would it be possible for the authors to include GEN as a baseline in Table 1? This addition would help readers better assess the relative performance enhancements introduced by GD-Net.**
>
> A4: We implemented the GEN [R2] model using the DGL package. The GEN model has 4 layers and a width of 64, the same as the implemented GD-Net. Experiment results show that GEN struggles with high training loss and consequently yields significantly lower-quality predictions compared to GD-Net and GCN:
>
> | Model  | #Params. | V. Err (Cover) | V. Err (Packing) |
> | ------ | -------- | -------------- | ---------------- |
> | GD-Net | 1,656    | 2.19E-6| 2.18E-4          |
> | GCN    | 34,306   | 1.81E-6        | 1.69E-4          |
> | GEN    | 18,177   | 0.021          | 0.024            |
> |||||
>
> Note that we have tuned hyper-parameters of GEN such as learning rate and dropout level, but GEN still under-performs GD-Net and GCN. This seems to contradict the results in [23] where GEN performs similarly or better than GCN. A potential reason for such discrepancy is that [23] applies GEN based on a tripartite modeling of LP, while our paper adopts the conventional bipartite modeling in [7].
>
> > **Q5: The paper highlights the unique advantage of unrolling methods in achieving exceptional size generalization. However, the current presentation in Table 2 might not fully capture this advantage. Would the authors consider augmenting their experimental setup to include:
> -Training on medium-scale datasets and testing on both small and large-scale datasets;
> -Introducing the performance of GCNs as baselines for a direct comparison;**
>
> A5: We train GD-Net on instances with 500 variables and generalize it to instances with 50 and 1,000 variables. The results are presented in the below table.
> ||||GD-Net|||GCN||
> |:-|-:|-:|-:|-:|-:|-:|-:|
> |Ins| #Vars.|Obj|A. Gap|R. Gap|Obj|A. Gap|R. Gap |
> |Cover|50|5.009|1.631|**48.30%**|5.027|1.649|48.80%|
> ||1000|6.186|2.788|83.60% |6.056|2.722|**81.70%**|
> |Packing|50|0.399|2.959|**88.10%**|0.331|3.028|90.20%|
> ||1000|3.001|0.333|**10.00%**|3.001|0.333|**10.00%**|
> |||||||||
>
> We see that while GD-Net requires much fewer parameters, it still achieves comparable or even better generalization as GCN does.
>
> The results might not be satisfactory for scenarios with huge differences in problem sizes. However, our method can still accelerate the solving of non-negative LPs where training and testing distributions are similar. To our knowledge, no existing work in L2O has tackled the generalization problem across such a huge size gap. Resolving this issue is an interesting research topic for future.
>
> [R1] Li Wang, Huaqing Wu, Wei Wang, and Kwang-Cheng Chen. Socially enabled wireless networks: Resource allocation via bipartite graph matching. IEEE Communications Magazine, 53(10):128–135, 2015.
>
> [R2] Guohao Li, et al. Deepergcn: All you need to train deeper GCNs, 2020.

---

> > ### Comment · Reviewer_h5Xh · 2024-08-08
> >
> > Thanks for your detailed response. The response has addressed my concerns and I have raised my rating to 7. This work establishes a link between the AK algorithm and GNNs and develops the lower bound matching GNNs for LP problems. This connection is not only innovative but also has the potential to spark interest and pave the way for future research in the field. For instance, it may inspire future GNN works in MILP, which is a more challenging and significant area.

---

> > > ### Author Response · Authors · 2024-08-09
> > >
> > > We greatly value your suggestions and recommendations. We will certainly incorporate the new results into our revised manuscript, as they not only strengthen our argument but also bridge the gap between theoretical bounds and practical applications.

---

### Official Review · Reviewer_hJjQ · 2024-07-11

**Soundness:** 2
**Presentation:** 4
**Contribution:** 1
**Rating:** 4
**Confidence:** 4

**Summary:**

This paper examines the expressive power of GNNs in representing linear programs (LPs). The authors first introduce a first-order iterative algorithm for packing and covering LPs, conceptualizing this algorithm as a GNN called GD-Net, applied to these LP types. They then provide a convergence rate of the proposed first-order algorithm, deriving a complexity upper bound for GNNs representing LPs. Finally, numerical experiments compare the performance of GD-Net with a general GCN.

**Strengths:**

The writing is clear and easy to follow. The mathematical aspects of the paper are technically correct.

**Weaknesses:**

The paper lacks significant contributions.

- The idea of conceptualizing iterative algorithms as GNNs is not novel:
  - Yang et al. "Graph Neural Networks Inspired by Classical Iterative Algorithms." ICML 2021.
  - Zhang and Zhao. "Towards Understanding Graph Neural Networks: An Algorithm Unrolling Perspective." 2022.

- The convergence rate of first-order algorithms for LPs is not new, even though GD-Net is not exactly the same with existing algorithms. For example:
  - Wang and Shroff. "A New Alternating Direction Method for Linear Programming." NeurIPS 2017.
  - Applegate et al. "Faster First-Order Primal-Dual Methods for Linear Programming using Restarts and Sharpness." Mathematical Programming 2023.

Note that these papers provide exact linear convergence rates, which are better than GD-Net's sublinear rate that only converges to an approximate solution. Additionally, the sublinear rate itself is not new, as it is provided by Awerbuch and Khandekar.

- The empirical performance of GD-Net is not convincing. The paper only compares GD-Net with GCN, without showing advantages over traditional algorithms like ADMM or PDHG. Furthermore, comparing with commercial LP solvers like CPLEX or Gurobi would be a more ambitious but necessary goal to demonstrate real-world applicability.

**Questions:**

Refer to "Weaknesses".

**Limitations:**

This theoretical paper appears to have no potential negative societal impact.

---

> ### Author Rebuttal · Authors · 2024-08-07
>
> > **Q1: The idea of conceptualizing iterative algorithms as GNNs is not novel. The convergence rate of first-order algorithms for LPs is not new. Given existing linear rates, the sublinear rate of GD-Net is not good enough.**
>
> Thanks for raising this insightful comment. We would like to make several clarifications on our contributions:
>
> (1) Our complexity bound of GD-Net focuses better **dependency (polylog) on problem sizes** $(m, n)$ rather than **better dependency on accuracy** $\epsilon$. (The latter is the focus of most convergence rate analysis of optimization algorithms.) In terms of dependency on problem sizes, our result improves the best-known bound of L2O networks **from polynomial to polylogarithmic**.
>
> We remark that the polylogarithmic dependency on $n$ is a little surprising, as it means that: For packing/covering LPs, which are **global optimization** problems, GD-Net can solve a near optimal solution based on **very local information**.
>
> (2) Our bound is established by unrolling a carefully chosen algorithm (i.e., AK algorithm). We are not aware of any other existing first-order algorithm such that unrolling it would lead to a polylog depth GNN.
>
> For example, consider the LP instance
> $$\max_{x\geq 0} \quad 1^T x$$
> $$\mbox{s.t.} \quad Ax \leq 1$$
> where A is an $n\times n$ Boolean matrix and contains three ones in each row and each column. Observe that the optimal solution is $(1/3,1/3,…,1/3)$. For any constant $\epsilon$, then the algorithm proposed by the paper “A New Alternating Direction Method for Linear Programming” needs $\Omega(n)$ iterations (since $R_x=\sup_k |x^k|=\Omega (n)$); in contrast, the GD-net only needs $\mathrm{polylog}(n)$ layers, exponentially better than $\Omega(n)$.
>
> ---------------------------------------------------
>
> > **Q2: The empirical performance of GD-Net is not convincing. The paper only compares GD-Net with GCN, without showing advantages over traditional algorithms like ADMM or PDHG. Furthermore, comparing with commercial LP solvers like CPLEX or Gurobi would be a more ambitious but necessary goal to demonstrate real-world applicability.**
>
> A2: We conduct experiments comparing GD-Net with PDLP [R1] and Gurobi. The table below presents the time taken by each method to achieve the same optimality to our GD-Net:
>
> | Ins.    | #Vars. |     Opt | GD-Net Obj | GD-Net time |   Gurobi |   PDLP |
> | :------ | -----: | ------: | ---------: | ----------: | -------: | -----: |
> |         |  1,000 |   3.334 |      3.701 |  **0.105s** |   0.244s | 0.919s |
> | Cover   |  5,000 | 100.667 |    130.931 |  **0.218s** |   9.401s | 0.921s |
> |         | 10,000 | 407.386 |    546.666 |  **0.335s** | 103.322s | 1.001s |
> |         |        |         |            |             |          |        |
> |         |  1,000 |   3.334 |      3.018 |  **0.095s** |   0.208s | 0.746s |
> | Packing |  5,000 |  100.88 |     78.994 |  **0.216s** |   3.980s | 0.756s |
> |         | 10,000 | 406.946 |     302.04 |  **0.314s** |   8.593s | 0.809s |
> |         |        |         |            |             |          |        |
>
> It shows that GD-Net consistently outperforms both Gurobi and PDLP in terms of speed while achieving comparable or superior objective values. More complementary experiments are presented in Author Rebuttal to bolster the robustness of our methods.
>
> [R1] David Applegate, et. al. Practical large-scale linear programming using primal-dual hybrid gradient, 2022.

---

> > ### Comment · Reviewer_hJjQ · 2024-08-12
> > **Response to authors**
> >
> > I sincerely appreciate the authors' efforts during the rebuttal stage, including their responses to my questions and the additional experiments provided.
> >
> > - The polylog complexity. I overlooked this aspect in my initial review. Thanks for pointing this out! I would like to tune my score based on this point.
> > - Novelty. As I mentioned earlier, the concept of unrolling a first-order method as a GNN is not new. The two papers I referenced are just examples on this topic. I strongly recommend that the authors thoroughly review the references within these papers, as well as those that cite them. A comprehensive discussion of the existing literature would be beneficial.
> > - Additional experiments. The results still appear unconvincing, as the "GD-Net Obj" is significantly higher than "Obj." Does this indicate that the objective/solution obtained by GD-Net is far from optimal? What are the accuracy (stopping tolerance) parameters for Gurobi and PDLP?

---

> > > ### Author Response · Authors · 2024-08-13
> > >
> > > Thanks for your valuable feedback.
> > >
> > > >  **Q1：Novelty. As I mentioned earlier, the concept of unrolling a first-order method as a GNN is not new. The two papers I referenced are just examples of this topic. I strongly recommend that the authors thoroughly review the references within these papers, as well as those that cite them. A comprehensive discussion of the existing literature would be beneficial.**
> > >
> > > **A1**: Thanks for the comment and the helpful advice.
> > >
> > > **Existing works on unrolling iterative algorithms as GNNs**: We acknowledge the reviewer’s point that unrolling first-order methods as GNNs has been explored in prior research. We will incorporate a comprehensive literature review in the revised version, as presented below. Please let us know if any other relevant works are missing from this review.
> > >
> > > > The design of GD-Net is based on the concept of unrolling iterative algorithms as GNNs. Indeed, there is a body of research that has explored this approach. For example, Velickovic et al. [R1] investigated solving basic graph problems (e.g., the shortest path, the minimum spanning tree) by GNNs. By unrolling classical graph algorithms (e.g., breadth-first search, Prim’s algorithm) as GNNs, they suggest that message-passing neural networks with a maximization aggregator may be best suited for such graph problems. Aiming at mitigating oversmoothing, long-range dependencies, and spurious edges issues of GNNs, Yang et al. [R2] proposed a new family of GNN layers by unrolling and integrating the update rules of two classical iterative algorithms, namely the proximal gradient descent and iterative reweighted least squares. References [R3, R4] showed that many existing GNN models (such as GCN, GAT, APPNP) can be viewed as unrolling gradient descent serving specific graph signal denoising problems. Chen et al. [R5] proposed new GNNs to improve graph signal denoising by unrolling sparse coding and trend ﬁltering algorithms. References [R6, R7, R8] bridge the gap between graph convolution and iterative algorithms by providing a unified optimization framework for GNNs.
> > >
> > > [R1] Petar Velickovic, et. al. Neural Execution of Graph Algorithms. ICLR 2020.
> > >
> > > [R2] Yongyi Yang, et. al. Graph Neural Networks Inspired by Classical Iterative Algorithms. ICML 2021.
> > >
> > > [R3] Yao Ma, et. al. A unified view on graph neural networks as graph signal denoising. CKIM 2021.
> > >
> > > [R4] Zepeng Zhang, and Ziping Zhao. Towards Understanding Graph Neural Networks: An Algorithm Unrolling Perspective. KDD 2022.
> > >
> > > [R5] Siheng Chen, et. al. Graph unrolling networks: Interpretable neural networks for graph signal denoising. IEEE Transactions on Signal Processing, 2021.
> > >
> > > [R6] Xuran Pan, et. al. A unified framework for convolution-based graph neural networks. Pattern Recognition, 2024.
> > >
> > > [R7] Meiqi Zhu, et. al. Interpreting and unifying graph neural networks with an optimization framework, Proceedings of the Web Conference, 2021.
> > >
> > > [R8] Hongwei Zhang, et. al. Revisiting graph convolutional network on semi-supervised node classification from an optimization perspective, *arXiv preprint arXiv:2009.11469* (2020).

---

> > > ### Author Response · Authors · 2024-08-13
> > >
> > > **Contribution of our paper**: Despite the above discussion, we would like to kindly provide further comment on our paper's contribution. Our main contribution is not merely "proposing another new GNN architecture by unrolling another iterative algorithm." Instead, our major contribution is twofold.
> > >
> > > 1. We present a new **theoretical explanation** of the empirical phenomenon that small-size GNNs can solve LPs. Specifically, we show that polylog-depth constant-width GNNs are expressive enough to solve a broad class of LPs. Moreover, **this polylog bound is also tight** (Please see Part 5: Tightness of Our Bound in "Author Rebuttal" for details).
> > > 2. For **practical use**, we propose a parameter-efficient and interpretable GNN architecture, namely GD-Net, for LPs. Experiments verifies its efficiency and effectiveness. Notably, GD-Net generates better solutions with an order of magnitude fewer parameters than GCN.
> > >
> > > We believe the above contribution is not straightforward because of the following reasons.
> > >
> > > 1. This work is motivated by observing a significant gap between empirical phenomenon and theoretical explanation in L2O (learning to optimize). Precisely, in practice, GNN with a modest width and fewer than ten layers often suffice to achieve good performance in approximating LP with hundreds of nodes and constraints. However, the best-known theoretical explanation [R9] requires the depth of GNN to grow polynomially with the problem size.
> > >
> > > 2) The selection of AK algorithm is nontrivial. AK-algorithm can be viewed as applying a variant of gradient descent on a potential function. The design of potential function is careful and nontrivial, since it is required to be differentiable and convex; and more importantly, any stationary point of the potential function should be a nearly optimal solution. These properties make AK algorithm has low computational complexity. We are not aware of any existing other first-order algorithm such that unrolling it would lead to a polylog depth GNN.
> > >
> > > [R9] Chendi Qian, et. al. Exploring the power of graph neural networks in solving linear optimization problems. AISTATS 2024.

---

> > > ### Author Response · Authors · 2024-08-13
> > >
> > > > **Q2: Additional experiments. The results still appear unconvincing, as the "GD-Net Obj" is significantly higher than "Obj." Does this indicate that the objective/solution obtained by GD-Net is far from optimal? What are the accuracy (stopping tolerance) parameters for Gurobi and PDLP?**
> > >
> > > **A2:** In our previous experiments,
> > >
> > > 1. The parameter $\epsilon$ in GD-Net is set to be 0.2, which means GD-Net is expected to output a $(1+\epsilon)=1.2$-approximation solution.
> > > 2. We run Gurobi and PDLP with default relative tolerance (precisely $10^{-6}$ for Gurobi, and $10^{-9}$ for PDLP), and stop them once they achieve the same objective value to the output of GD-Net.
> > >
> > > We acknowledge that, in general, an L2O network may not achieve the very high precision of $10^{-6}$ or $10^{-9}$ that traditional optimization algorithms can reach. This is because traditional algorithms can run for an arbitrary number of iterations, given sufficient resources, while the layers of a neural network are fixed after training.
> > >
> > > **However, L2O networks can still help accelerate the solving of optimization problems that require higher precision.** Specifically, as demonstrated in many related studies [R10, R11, R12], the L2O approach is often used to quickly generate feasible solutions or to provide strong initial warm-starts for traditional optimization algorithms. **Combining L2O networks with traditional optimization algorithms can achieve the desired precision faster than using traditional optimization algorithms alone.**
> > >
> > > To demonstrate the practical utility of GD-Net, we conducted experiments to assess how effectively GD-Net can serve as a warm start to enhance the efficiency of PDLP. We measured the time it takes to solve problems via PDLP using warm starts provided by GD-Net, aiming to meet predefined stopping criteria. The relative tolerance was set at 1e-3, which is a typical precision requirement in real-world applications. We also recorded the time PDLP required to resolve instances for comparison. Additionally, we calculated the improvement ratio by comparing the time taken by PDLP alone with the time taken when using GD-Net's output as a warmstart.
> > >
> > > |dataset|size|time for PDLP alone (s)|time for GD-Net + PDLP (s)|Improvement Ratio|
> > > |-:|-:|-:|-:|-:|
> > > |Packing|5000|5.461|4.920|9.906%|
> > > |Covering|5000|6.893|5.619|18.483%|
> > > ||||||
> > >
> > > The experimental results show that utilizing GD-Net as warm-starts can reduce the time of solving by about 10% and 18% respectively for Packing and Covering dataset, comapred to using PDLP alone.
> > >
> > > We thank the reviewer again for the valuable comments and suggestions. We will include the above results and discussions in the manuscript.
> > >
> > > [R10] Qinyu Han et al. A GNN-guided predict-and-search framework for mixed-integer linear programming. ICLR, 2023.
> > >
> > > [R11] Rajiv Sambharya et al. Learning to Warm-Start Fixed-Point Optimization Algorithms. JMLR, 2024.
> > >
> > > [R12] Rajiv Sambharya et al. End-to-end learning to warm-start for real-time quadratic optimization. Learning for Dynamics and Control Conference, 2023.

---

### Official Review · Reviewer_xnS6 · 2024-07-13

**Soundness:** 4
**Presentation:** 3
**Contribution:** 4
**Rating:** 7
**Confidence:** 2

**Summary:**

The paper investigates the capability of small-sized Graph Neural Networks (GNNs) to solve linear programming (LP) problems, specifically focusing on polylogarithmic-depth, constant-width GNNs. It provides both theoretical proofs and empirical evidence demonstrating that these GNN architectures can effectively solve packing and covering LPs, which are common in various optimization contexts. The introduction of a novel GNN architecture, termed GD-Net, is highlighted, showing superior performance in terms of parameter efficiency and problem-solving capability compared to traditional GNNs.

**Strengths:**

1. **Theoretical Foundation**: The paper successfully bridges the gap between theoretical predictions and empirical results by proving that small-sized GNNs can efficiently solve packing and covering LPs.
2. **Innovative Architecture**: The introduction of GD-Net, which significantly outperforms existing GNN models using fewer parameters, provides practical insights into the design of efficient neural network architectures for optimization problems.
3. **Comprehensive Experiments**: Extensive empirical evaluations demonstrate the effectiveness of the proposed approach across different datasets and problem sizes, which solidifies the paper’s claims.
4. **Impactful**: As you noted, the results are inspiring and provide valuable guidance for the L2O community on GNN structure and size design.

**Weaknesses:**

**Scope of Applicability**: The paper primarily focuses on packing and covering LPs. The generalizability of the proposed methods to other types of LPs or optimization problems is not addressed.

**Questions:**

1. Could the authors elaborate on potential modifications or extensions of the GD-Net architecture that might enable it to handle a broader range of LPs or even mixed integer linear programming problems?
2. What are the limitations in terms of scalability when dealing with extremely large datasets or more complex network architectures?

**Limitations:**

**Generalizability**: The techniques are validated primarily on packing and covering LPs, and there is limited discussion on their effectiveness for other classes of LPs or more complex optimization tasks.

Although I am not an expert in this specific area, the encouraging results give the L2O community some guidance on designing GNN architectures and sizes. I defer to other reviewers for a more detailed critique.

---

> ### Author Rebuttal · Authors · 2024-08-07
>
> > **Q1: Could the authors elaborate on potential modifications or extensions of the GD-Net architecture that might enable it to handle a broader range of LPs or even mixed integer linear programming problems?**
>
> A1: Intuitively, GD-Nets can be viewed as unrolling the gradient descent on a carefully selected potential function. This potential function is differentiable and convex; more importantly, any stationary point of the potential function is a nearly optimal solution. For a broader class of LPs and even MILPs, a natural potential modification of the GD-net is to design another potential function that still enjoys the above good properties.
>
> ---
> > **Q2: What are the limitations in terms of scalability when dealing with extremely large datasets or more complex network architectures?**
>
> A2: If we understand the question correctly (please tell us if not so), "extremely large datasets" means datasets on large problem dimensions, say, LPs with 10,000 variables or above. When dealing with such datasets, we need complex network architectures with more parameters. There are two main limitations in terms of scalability that need to be addressed.
>
> **Computational resources**: Handling large datasets requires longer training time, a larger amount of memory, and higher costs for infrastructure. Systems with limited resources would struggle to handle large datasets, resulting in performance issues and inefficiencies. In that sense, designing more parameter-efficient L2O networks is crucial for scaling up and enhancing performance in resource-constrained systems. Our GD-Net, for example, offers an improvement in parameter efficiency compared to traditional GCNs. Besides, efficient training and inference methods are also needed to reduce computational costs.
>
> **Number of training examples**: As the problem dimension grows and the network becomes more complex (with more parameters), the risk of overfitting (a.k.a. poor generalization) also increases. To mitigate overfitting, the number of training examples should grow along with the problem dimension and model parameters. This post a challenge on data collection. Efficient data generation methods or self-supervised training paradigms can be leveraged to resolve this challenge.
>
> Thanks for the inspiring comments. We will add the above discussion on Q1 and Q2 in the next version.

---

> > ### Comment · Reviewer_xnS6 · 2024-08-08
> >
> > Thank you for your detailed and thoughtful rebuttal. I appreciate the insights you've provided on extending the GD-Net architecture to handle a broader range of LPs and potentially MILPs, as well as the considerations for scalability with large datasets and complex network architectures. I'm genuinely excited to see how your work progresses and its potential application in the MILP domain. I will maintain my score, and I look forward to the future developments of your research.

---

> > > ### Author Response · Authors · 2024-08-09
> > >
> > > Thank you for your encouraging and insightful feedback. We greatly appreciate your recognition of our efforts. Your insights are valuable as we advance our work. We will definitely incorporate these new results into our revised manuscript to further enhance our work.

---

### Author Rebuttal · Authors · 2024-08-07

We sincerely thank the reviewers for their valuable comments. We provide detailed responses individually to each reviewer. Note that more than one reviewer suggested additional numerical experiments to bolster the robustness of our findings. We conduct these experiments and summarize as below.

---
**1. Comparison with Traditional Algorithms and Commercial Solvers:**

We include two additional baselines: a traditional algorithm PDLP [R2] and a commercial solver Gurobi. The table below presents the time taken by each method to achieve the same optimality to our GD-Net:

| Ins.    | #Vars. |     Opt | GD-Net Obj | GD-Net time |   Gurobi |   PDLP |
| :------ | -----: | ------: | ---------: | ----------: | -------: | -----: |
|         |  1,000 |   3.334 |      3.701 |  **0.105s** |   0.244s | 0.919s |
| Cover   |  5,000 | 100.667 |    130.931 |  **0.218s** |   9.401s | 0.921s |
|         | 10,000 | 407.386 |    546.666 |  **0.335s** | 103.322s | 1.001s |
|         |        |         |            |             |          |        |
|         |  1,000 |   3.334 |      3.018 |  **0.095s** |   0.208s | 0.746s |
| Packing |  5,000 |  100.88 |     78.994 |  **0.216s** |   3.980s | 0.756s |
|         | 10,000 | 406.946 |     302.04 |  **0.314s** |   8.593s | 0.809s |
|         |        |         |            |             |          |        |

GD-Net consistently outperforms both Gurobi and PDLP in terms of speed while achieving comparable or superior objective values.

---
**2.Comparison with GEN [R3]**

We implemented the GEN model using the DGL package. The GEN model has 4 layers and a width of 64, the same as the implemented GD-Net. Experiment results show that GEN struggles with high training loss and consequently yields significantly lower-quality predictions compared to GD-Net and GCN:

| Model  | #Params. | V. Err (Cover) | V. Err (Packing) |
| ------ | -------- | -------------- | ---------------- |
| GD-Net | 1,656    | 2.19E-6        | 2.18E-4          |
| GCN    | 34,306   | 1.81E-6        | 1.69E-4          |
| GEN    | 18,177   | 0.021          | 0.024            |
|        |          |                |                  |

---
**3. More Practical Setting**

We consider the Bipartite Maxflow, a common model formulation applied to areas such as wireless communication [R1]. In our dataset, each bipartite graph is obtained by deleting all edges between $V'$ and $U'$ from a fully connected bipartite graph, and then randomly sample from the remaining edges with a probability of 60%, where $V'$ (and $U'$ resp.) is a random subset consisting of half of the left nodes (and the right nodes).

||| GD-Net ||| GCN     |        |
| ------ |-|-|-|-|-|- |
| #Nodes | Obj      | A. Gap | R. Gap    | Obj   | A. Gap  | R. Gap |
| 1200   | 35398.62 | 429.8  | **1.20%** | 31206 | 4622.41 | 12.89% |
| 2000   | 58844.8  | 943.33 | **1.58%** | 52085 | 7703.14 | 12.88% |
||||||||

According to the result, GD-Net consistently obtains better prediction compared to GCN, with only 1% optimality gap from optimal solutions.

We conducted comparisons against the Ford Fulkerson heuristic, specifically designed for solving maximum network flow problems. The table below presents the time taken to achieve the same optimality to GD-Net:

| #Nodes | GD-Net Obj | GD-Net time | Ford Fulkerson time |
| -: | -: | -: | -: |
|1200 |35398.62 |0.592s |2.152s |
|2000 |58844.80 |1.691s |9.184s |
|||||

GD-Net is significantly faster than the Ford Fulkerson heuristic in achieving high-quality solutions, demonstrating its efficiency and effectiveness.


---
**4. Size Generalization**

We train GD-Net on instances with 500 variables and generalize it to instances with 50 and 1,000 variables. The results are presented in the below table.

|         |          |       | GD-Net |            |       |    GCN |            |
| :------ | -------: | ----: | -----: | ---------: | ----: | -----: | ---------: |
| Ins     | #Vars. |   Obj | A. Gap |     R. Gap |   Obj | A. Gap |     R. Gap |
| Cover   |       50 | 5.009 |  1.631 | **48.30%** | 5.027 |  1.649 |     48.80% |
|         |     1000 | 6.186 |  2.788 |     83.60% | 6.056 |  2.722 | **81.70%** |
| Packing |       50 | 0.399 |  2.959 | **88.10%** | 0.331 |  3.028 |     90.20% |
|         |     1000 | 3.001 |  0.333 | **10.00%** | 3.001 |  0.333 | **10.00%** |
|         |          |       |        |            |       |        |            |

We see that the GD-Net while requiring much fewer parameters, still achieves comparable or even better generalization as GCN does. To the best of our knowledge, no existing work in L2O has tackled the generalization across such a huge size gap. Improving the large-gap size generalization is an interesting research topic for future research.


---
**5. Tightness of Our Bound**

Our main result says that “polylogarithmic-depth constant-width GNNs are expressive enough to solve packing/covering LPs." Here, we remark that the polylogarithmic dependency on depth is also necessary.

Specifically, Kuhn et al. showed that (first paragraph on page 6 in [R4]): For the fractional maximum matching problem, a special kind of packing LP, every constant-factor approximation distributed algorithm requires at least $\Omega(\sqrt{\log n/\log\log n})$ rounds. Moreover, since one layer of GNNs can be naturally simulated by one round of distributed LP algorithms (see the second paragraph on page 5 in [R4]), we conclude that GNNs need at least $\Omega(\sqrt{\log n/\log\log n})$ layers.

[R1] Li Wang, Huaqing Wu, Wei Wang, and Kwang-Cheng Chen. Socially enabled wireless networks: Resource allocation via bipartite graph matching. IEEE Communications Magazine, 53(10):128–135, 2015.

[R2] David Applegate, et. al. Practical large-scale linear programming using primal-dual hybrid gradient, 2022.

[R3] Guohao Li, et al. Deepergcn: All you need to train deeper GCNs, 2020.

[R4] Fabian Kuhn, et. al. Local Computation: Lower and Upper Bounds. J. ACM 63(2): 17:1-17:44 (2016).

---

### Decision · Program_Chairs · 2024-09-25

**Decision:**

Accept (poster)

**Comment:**

The paper studies the effectiveness of GNNs for solving LPs, through connections with a variant of gradient descent on specific potential functions. The reviewers found the proposed method effective and of interest to the community.